# DNA methylation loss promotes immune evasion of tumours with high mutation and copy number load

Hyunchul Jung [1,10], Hong Sook Kim[2,10], Jeong Yeon Kim[1], Jong-Mu Sun[2], Jin Seok Ahn[2], Myung-Ju Ahn[2], Keunchil Park[2], Manel Esteller[3,4,5,6,7], Se-Hoon Lee[2,8] & Jung Kyoon Choi[1,9]

Mitotic cell division increases tumour mutation burden and copy number load, predictive markers of the clinical benefit of immunotherapy. Cell division correlates also with genomic demethylation involving methylation loss in late-replicating partial methylation domains. Here we find that immunomodulatory pathway genes are concentrated in these domains and transcriptionally repressed in demethylated tumours with CpG island promoter hypermethylation. Global methylation loss correlated with immune evasion signatures independently of mutation burden and aneuploidy. Methylome data of our cohort ($n = 60$) and a published cohort ($n = 81$) in lung cancer and a melanoma cohort ($n = 40$) consistently demonstrated that genomic methylation alterations counteract the contribution of high mutation burden and increase immunotherapeutic resistance. Higher predictive power was observed for methylation loss than mutation burden. We also found that genomic hypomethylation correlates with the immune escape signatures of aneuploid tumours. Hence, DNA methylation alterations implicate epigenetic modulation in precision immunotherapy.

[1] Department of Bio and Brain Engineering, KAIST, Daejeon 34141, Republic of Korea. [2] Division of Hematology/Oncology, Department of Medicine, Samsung Medical Center, Sungkyunkwan University School of Medicine, Seoul 06351, Republic of Korea. [3] Cancer Epigenetics and Biology Program (PEBC), Bellvitge Biomedical Research Institute (IDIBELL), L'Hospitalet, Barcelona, Catalonia, Spain. [4] Centro de Investigacion Biomedica en Red Cancer (CIBERONC), 28029 Madrid, Spain. [5] Institucio Catalana de Recerca i Estudis Avançats (ICREA), Barcelona, Catalonia, Spain. [6] Physiological Sciences Department, School of Medicine and Health Sciences, University of Barcelona (UB), Barcelona, Catalonia, Spain. [7] Josep Carreras Leukaemia Research Institute (IJC), Badalona, Barcelona, Catalonia, Spain. [8] Department of Health Sciences and Technology, Samsung Advanced Institute of Health Science and Technology, Sungkyunkwan University, Seoul 06351, Republic of Korea. [9] Penta Medix Co., Ltd., Seongnam-si, Gyeongi-do 13449, Republic of Korea. [10]These authors contributed equally: Hyunchul Jung, Hong Sook Kim. Correspondence and requests for materials should be addressed to S.-H.L. (email: shlee119@skku.edu) or to J.K.C. (email: jungkyoon@kaist.ac.kr)

Cancer immunotherapy based on checkpoint blockade has become highly effective in a subset of patients with different types of human cancers. In particular, antibody-mediated interventions targeting cytotoxic T lymphocyte antigen-4 (CTLA-4) and programmed death receptor-1 (PD-1) on T lymphocytes and the principal ligand (PD-L1) on tumour cells can reverse tumour-induced immunosuppression and induce durable clinical responses[1].

A major challenge facing current immunotherapies is the identification of biomarkers that predict clinical responses to CTLA-4 and PD-1/PD-L1 blockade. Overall, the mutational or neoantigen load[2–5] and pre-existing T cell infiltration[6,7] are indicators of clinical benefit of checkpoint blockade. On the other hand, somatic copy number alterations (SCNAs)[8–10], tumour heterogeneity[11], and the genetic alteration of specific genes[12] or pathways[13] have been identified as resistance factors.

Tumour cells produce neoantigens or antigens that the immune system never encountered without cancer. The epitopes of neoantigens are displayed on the surface of cancer cells and provoke immune response. Therefore, tumours with high mutation load are more likely to respond to anti-immunosuppressive strategies based on checkpoint blockade[2–5]. Mutation load increases as a result of replicative errors during cell division. Not only mutations but also methylation losses accumulate during successive rounds of cell division[14]. Global hypomethylation and CGI hypermethylation represent the hallmark methylation changes in cancer[15]. Considering its association with late replication timing, progressive methylation loss may occur due to the failure of methylation maintenance machinery to remethylate newly synthesized daughter stands during DNA replication[14,16]. However, the influence of genomic methylation loss through cell divisions has never been investigated in the context of cancer immunotherapy.

Meanwhile, SCNAs emerged as a resistance parameter[8–10]. A pan-cancer analysis discovered the association of SCNAs with molecular signatures of cytotoxic immune activity across diverse tumour types[10]. Particularly, highly aneuploid tumours with extensive chromosome- or arm-level SCNAs showed a lower expression of markers indicating infiltrating immune cells. In contrast, focal SCNAs mainly correlated with cell proliferation markers instead of immune activity signatures. However, the mechanism by which aneuploidy affects immune cell infiltration remains unknown. Global demethylation in cancer promotes chromosomal instability[17–20], particularly involving large-scale alterations leading to aneuploidy[21–23]. Therefore, we investigated the relationships between methylation changes and aneuploidy.

Here, we performed large-scale systematic analyses of the molecular data of TCGA samples across a variety of tumour types. We examined the relationships of global methylation levels with markers of cell proliferation, mutation burden, SCNA levels, markers of infiltrating immune cells, and activity of immune-response genes. Importantly, we tested our hypotheses developed from the molecular analyses by using our lung cancer cohort. This is the first study that inspected DNA methylation patterns in the molecular and clinical data with regards to cancer immunotherapy. As a result, we suggest that as an important predictive marker in immunotherapy, genomic demethylation implicates epigenetic modulation as a combination regimen for precision immunotherapy.

## Results

**Global methylation correlates with immune signatures**. Our pan-cancer analyses of TCGA data demonstrated that markers of cell proliferation tightly correlate with mutation burden and aneuploidy across cancer types and among samples within each cancer type (Supplementary Fig. 1). Our measure of genomic demethylation based on long interspersed nuclear element-1 (LINE-1 or L1)[24,25] probes (Supplementary Fig. 2) also strongly correlated with cell proliferation markers (Fig. 1a and Supplementary Data 1). Global methylation loss was also associated with an increase in mutation burden (Fig. 1b) and chromosomal SCNA load (Fig. 1c), two types of genomic aberrations that accumulate through cell division (Supplementary Fig. 5).

Notably, we found a correlation between the global L1 methylation levels and immune signatures such as markers of tumour-infiltrating CD8 + T cells (Fig. 1d, Supplementary Data 1, and Supplementary Fig. 3). However, immune cell markers are expected to correlate with mutation burden and also are known to be associated with aneuploidy[10]. To disentangle this intercorrelation, we performed multiple regression of the expression level of each gene on sample-level features, namely, global L1 methylation, mutation burden, aneuploidy, tumour purity, age, and tumour stage. In this manner, we were able to determine that immune infiltrates are associated with the global methylation levels independently of mutation burden and aneuploidy when purity, age, and tumour stage are adjusted (Fig. 2a).

Significant correlations with genomic demethylation were observed also for immunomodulatory pathways that should include genes expressed in tumour cells. These include antigen processing and presentation, major histocompatibility complex (MHC), cytokine–cytokine receptor interaction, interferon or other cytokine signaling, and complement and coagulation (Fig. 2b). There is an emerging role for the complement system in regulating the antitumour immune response[26]. The correlation of the cell proliferation markers was in the opposite direction to that of the immune cell markers or immunomodulatory genes (Fig. 2b). We confirmed that the global L1 methylation level itself was not affected by the leukocyte fraction, which correlated only with immune gene expression (Supplementary Fig. 4).

**Repression of immune genes in late-replicating regions**. To focus on the repression of immune-response genes in tumour cells, we excluded genes that are specifically expressed in the immune system from the following analyses. Because methylation loss occurs primarily in late-replicating regions[14,16], we examined whether the transcriptional activity of late-replicated genes are affected in the tumours that underwent global demethylation. By using cell line data, we identified genes that are replicated earlier or later in cancer compared with normal cells (Supplementary Data 2). As a result, we found that the genes replicating late in cancer were significantly repressed in the demethylated tumours (Fig. 3a) with CpG island (CGI) promoter hypermethylation (Fig. 3b). In contrast, the early-replicating genes tended to be overexpressed in the demethylated tumours (Fig. 3c).

Overall, immune-related pathways were overrepresented in the late-replicating regions while cell cycle genes were concentrated in the early-replicating regions (Fig. 3d). More specifically, the pathways most enriched for the late-replicating genes in cancer included cytokine-cytokine receptor interaction, interferon-α/β (IFN-α/β) signaling, and RIG-1/MDA5-mediated IFN-α/β induction (Fig. 3e and Supplementary Table 1). RIG-1/MDA5-mediated induction of IFN-α/β represents innate immune reaction against RNA viruses. In contrast to our data from tumour-intrinsic demethylation, treatment of methylation inhibitors was shown to induce double-stranded RNAs (dsRNAs) derived from endogenous retroviruses (ERVs) and LINEs, resulting in the activation of the IFN-α/β response in cancer[27–29]. Without the silencing of the IFN-α/β pathway, genomic demethylation would cause the antiviral response and facilitate antitumour immune reaction as demonstrated with demethylating agents. We measured the

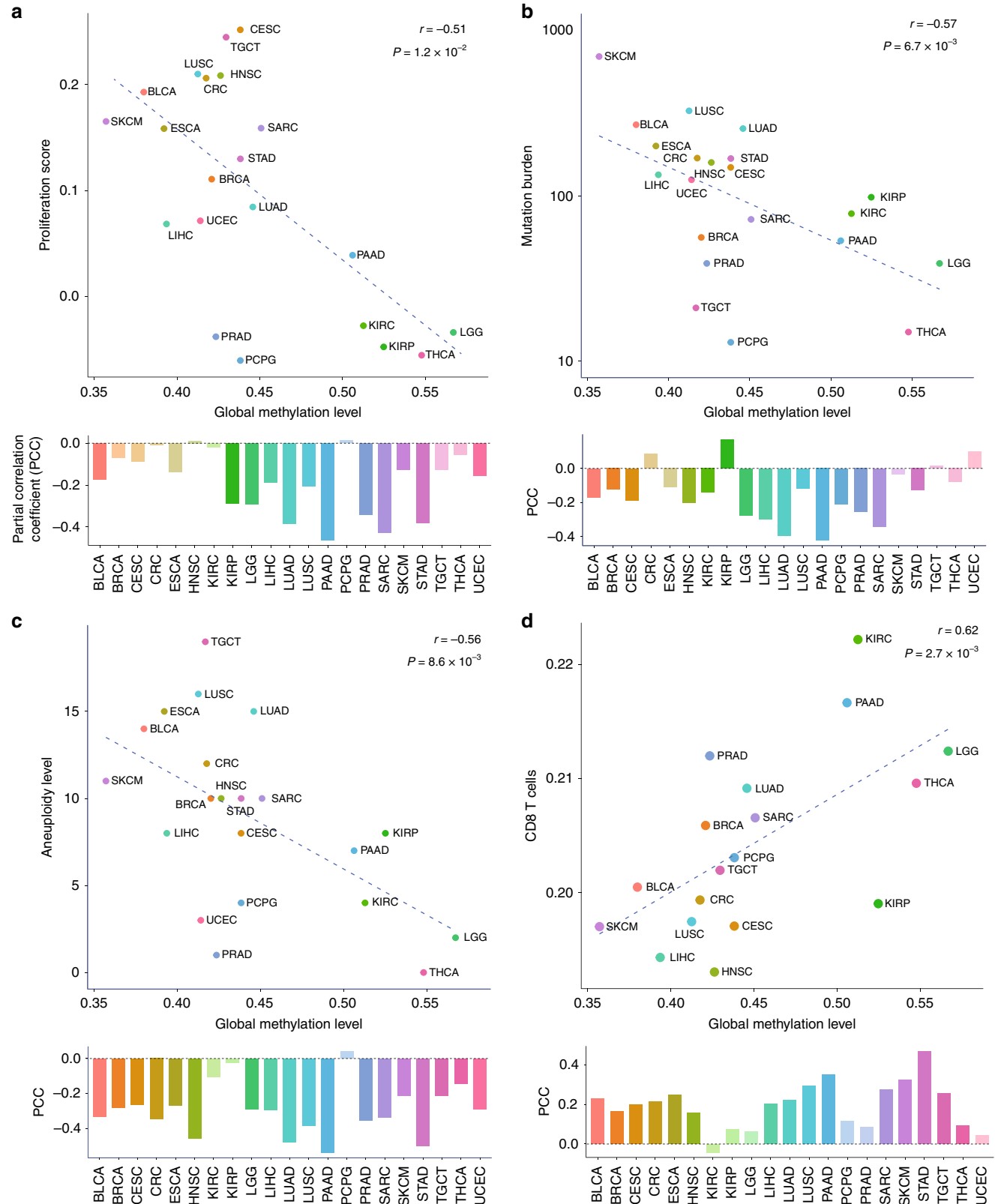

**Fig. 1** Correlates of global DNA methylation level. Correlation between genomic methylation levels and cell proliferation markers **a**, mutation burden **b**, aneuploidy level **c**, and tumour-infiltrating CD8 + T cell markers **d** across and within 21 cancer types. **a–d** The median values were obtained for each cancer type, and statistical significance was evaluated using Spearman's correlation (upper scatterplots). Three outlier cancer types (ESCA, STAD, and UCEC) are not shown for the CD8 + T cell correlation, but included when evaluating Spearman's correlation. For the correlation within each cancer type (lower bar graphs), Spearman's partial correlation was used to adjust for tumour purity. Tumour types showing significant partial correlation coefficient ($P < 0.05$) were shaded in darker colours

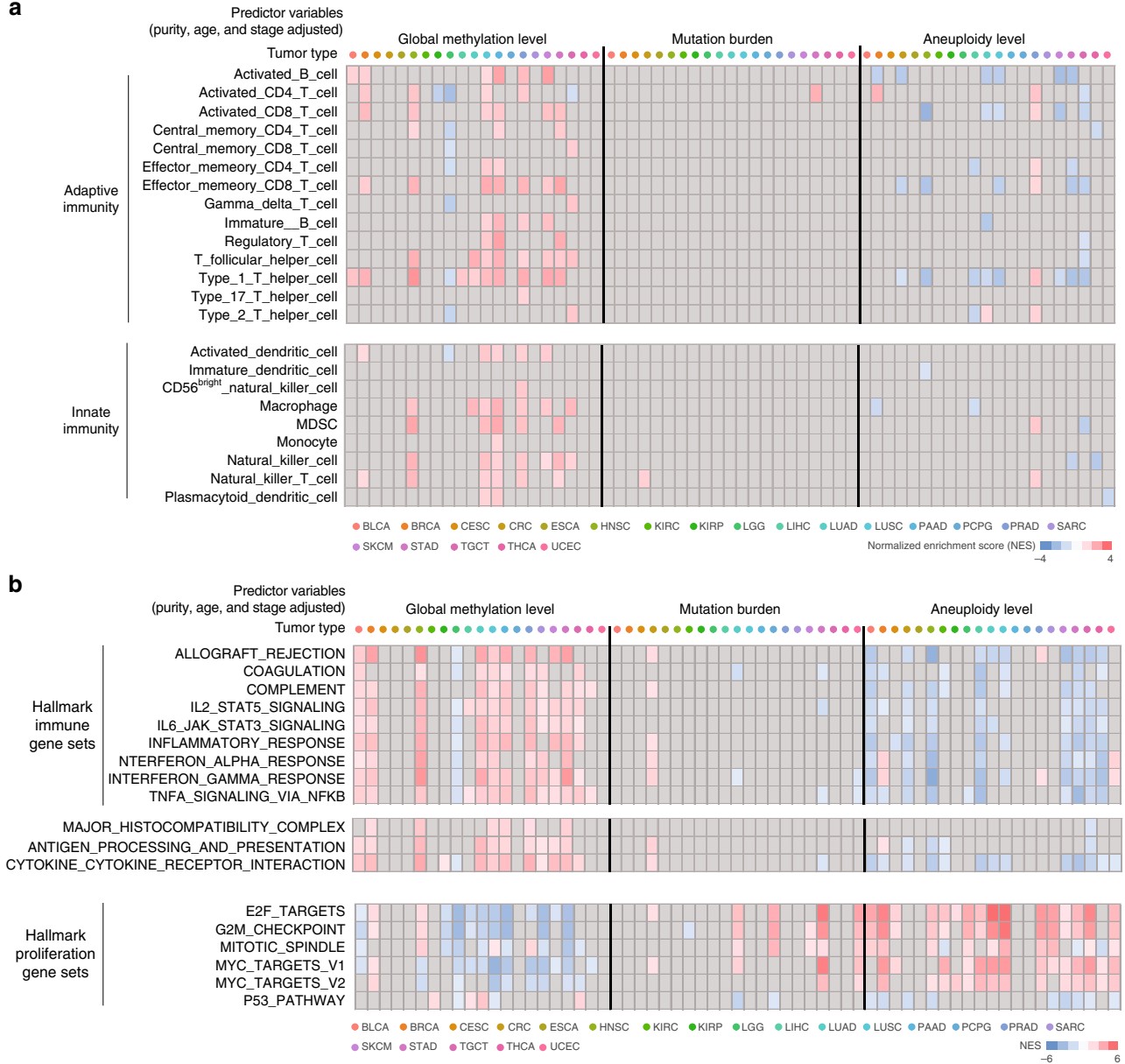

**Fig. 2** Genomic methylation loss correlates with immune evasion signatures. **a** Heatmap of Gene Set Enrichment Analysis (GSEA)[48] normalized enrichment scores (NESs) for gene sets representing various types of innate and adaptive immune cells (rows). For each gene per cancer type, a linear regression model was fit using mRNA expression level as response variable and global methylation, mutation burden, aneuploidy, tumour purity, age, and tumour stage as predictors. For each of three predictors (global methylation level, mutation burden, aneuploidy level), GSEA was performed on genes with significant regression coefficients. Cells with significant NES (FDR <0.25) are colour-scaled. **b** Heatmap of GSEA NESs for hallmark immune and proliferation gene sets and genes involved in antigen processing and presentation, MHC, or cytokine-cytokine receptor interaction

expression levels of ERVs and L1s from the tumour samples. The correlation of their expression levels with the indicators of cytotoxic immune activity was not positive but negative in general (Supplementary Fig. 5), implying that IFN-α/β silencing overrides the immune-stimulatory effects of ERV/L1 expression by genomic demethylation.

**Repression of immune genes in partial methylation domains**. Methylation loss in late-replicating regions engages the formation of heterochromatic structure termed partial methylation domains (PMDs) as opposed to highly methylated domains (HMDs)[16]. PMDs were first discovered as contiguous regions with lower levels of CpG methylation in differentiated cells[30]. PMD-like

long-range tumour demethylation was discovered in colon[31], breast[32], and brain[33] cancers. A recent study showed PMD demethylation is a common feature of diverse cancer type[16]. Such long-range demethylation in cancer is accompanied with gene silencing programs. Genes within PMDs in differentiated cells are under-expressed[30]. Similarly, genes in PMDs in various types of cancers are largely silenced by the formation of repressive chromatin structures or via CGI hypermethylation[31–35].

PMDs were characterized by and defined based on the high variability of solo-WCGW methylation levels across samples[16]. Our inspection of the methylation variability and replication timing of various PMDs led to three distinct subclasses (Fig. 4a). In accordance with a previous report[36], the properties of PMDs were associated with their genomic length with shorter PMDs

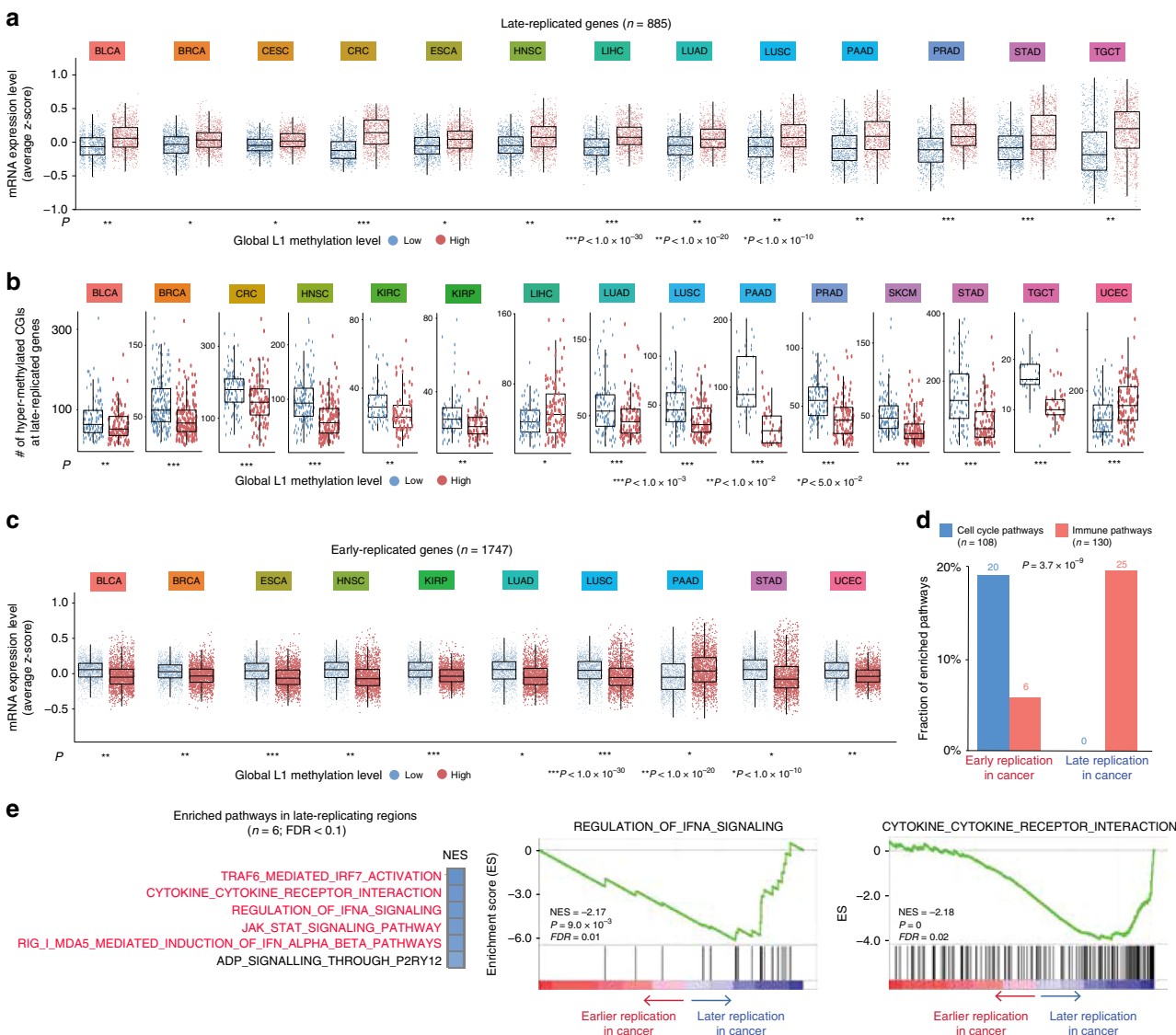

**Fig. 3** Characterization of genes in late-replicating regions. **a** Comparison of the expression levels of late-replicated genes between tumours with low and high global methylation. Tumour types for which the paired $t$-test $P < 1 \times 10^{-10}$ are shown. **b** Comparison of the number of hypermethylated CpG island promoters of late-replicating genes between tumours with low and high global methylation. Tumour types with $P < 0.05$ (two-sided Mann–Whitney U test) are shown. In the boxplots, the centre line, bounds of box, and whiskers represent the 50th, 25th and 75th, and 5th, and 95th percentiles, respectively. **c** Comparison of the expression levels of early-replicated genes between tumours with low and high global methylation. Tumour types for which the paired t-test $P < 1 \times 10^{-10}$ are shown. **d** Fraction of enriched cell-cycle and immune pathways according to replication timing. The number of the pathways showing significant enrichment (FDR <0.25) by GSEA is indicated above the bars. **e** Significantly enriched pathways for genes in late-replicating regions and the two representative GSEA plots. Genes that are specifically expressed in the immune system were excluded

characterized by earlier replication timing (Fig. 4a, b). Strikingly, immunomodulatory pathway genes involved in antigen processing and presentation, cytokine-cytokine receptor interaction, and JAK-STAT signaling pathway were concentrated in the short PMDs (Fig. 4c, d and Supplementary Table 2). The INF-α family genes were in the short PMDs (Fig. 4e). Also, 8 HLA genes, including HLA-DQA1, HLA-DRA, and HLA-DRB1, were located within the short PMDs. Consistent with the late-replicating regions (Fig. 3a, b), the short PMDs were accompanied with gene repression (Fig. 4f) and CGI hypermethylation (Fig. 4g) in demethylated tumours. Hypermethylated CGIs are most abundant within 150 kb of PMD boundaries[31]. The enriched immune genes (Fig. 4c, d) were significantly concentrated near PMD boundaries with the average distance of 143 kb (Fig. 4h), suggesting that these genes are particularly prone to promoter methylation.

**Global methylation predicts responses to immunotherapy.** To test whether global methylation alterations affect the clinical benefit of immunotherapy, we generated methylome and exome data for 60 samples in an anti-PD-1/PD-L1 cohort in lung cancer collected from Samsung Medical Center (SMC) (Supplementary Table 3). Also, we employed an additional anti-PD-1 lung cancer cohort composed of 81 methylomes and 22 exomes from Bellvitge Biomedical Research Institute (IDIBELL)[37]. For validation, we utilized data from 40 TCGA melanoma patients who received immunotherapies. The summary of the three cohorts is provided in Fig. 5a.

The samples from the combined lung cancer cohort were divided into global low versus high methylation groups according to the L1 methylation levels. The global low methylation group exhibited decreased genomic (open sea/shelf) methylation and increased CGI/shore methylation (Fig. 5b, c). In agreement with

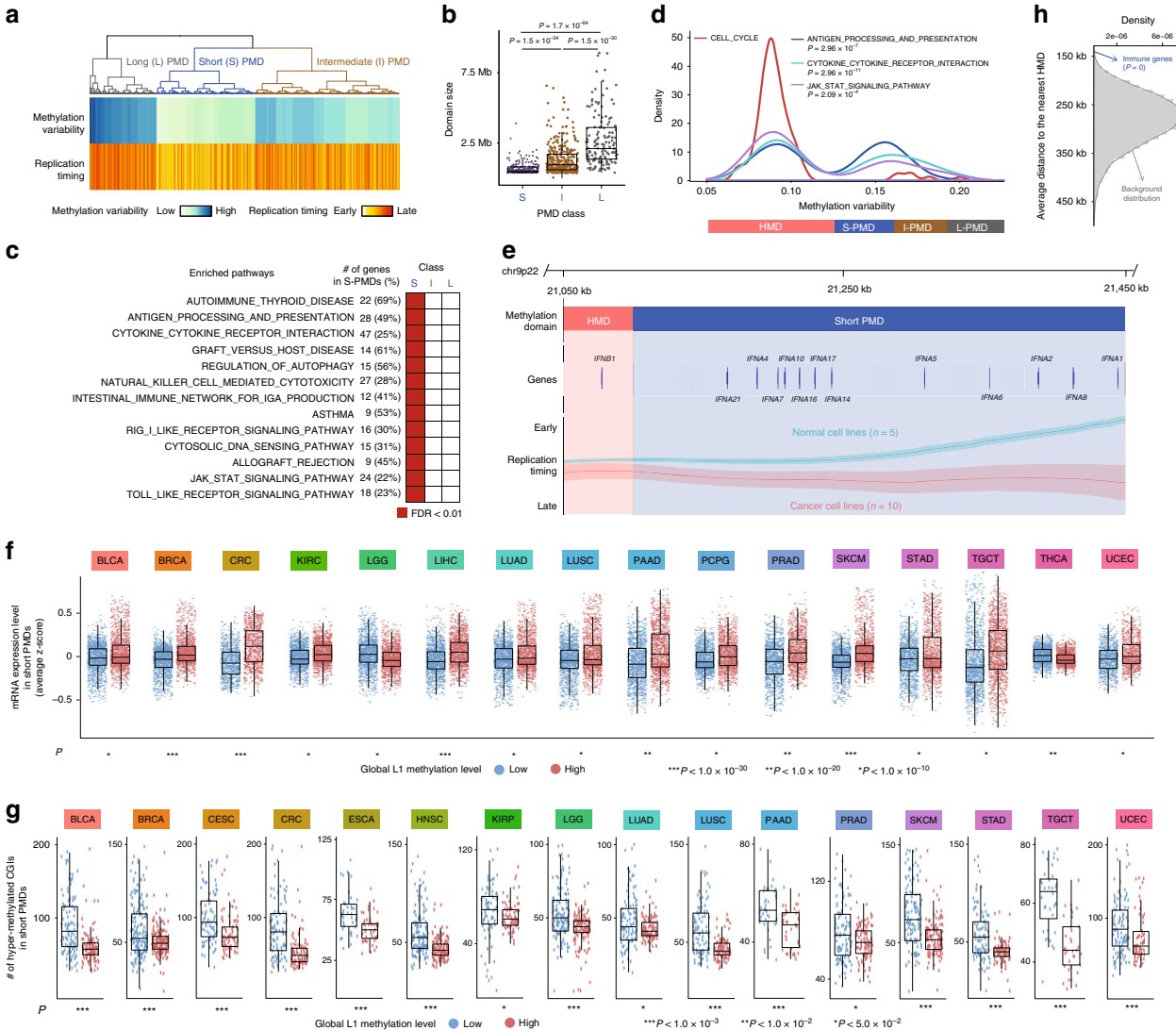

**Fig. 4** Characterization of genes in partially methylated domains. **a** Identification of PMD subclasses through hierarchical clustering on methylation variability and replication timing. **b** Comparison of PMD size between the identified PMD subclasses. **c** Enrichment of genes in immune-related pathways in the short PMDs. **d** Distribution of domain methylation variability for genes in cell cycle and immune pathways. The two-sample Kolmogorov-Smirnov test was used to assess deviation from the distribution of the cell cycle genes. **e** IFN-α genes in a short PMD with late replication timing. The mean and standard error of the weighted average signals of replication timing in normal cells and cancer cells are shown. **f** Comparison of the mRNA expression level of genes in the short PMDs between tumour samples with low and high global methylation. Tumour types for which the paired t-test $P < 1.0 \times 10^{-10}$ are shown. **g** Comparison of the number of hyper-methylated promoter CGIs in the short PMDs between tumour samples with low and high global methylation. Tumour types with $P < 5.0 \times 10^{-2}$ (two-sided Mann–Whitney U test) are shown. **h** Concentration of immune genes near PMD boundaries. The average distance of the immune-related genes (from b) to the nearest HMDs is marked by an arrow. The statistical significance of the observed average distance was assessed based on a null distribution generated by using random PMD genes

our pan-cancer molecular data analyses, the global low methylation samples showed high mutation burden and aneuploidy as well as CGI hypermethylation in the short PMDs (Fig. 5d). Transcriptome data of the SMC cohort and TCGA cohort showed that genes involved in the MHC and cytokine-cytokine receptor interaction were significantly enriched for repression in globally demethylated tumours of both cohorts (Supplementary Fig. 6 and Supplementary Data 3).

High mutation load is associated with the clinical benefit of checkpoint blockade[2–5]. However, when we stratified patient samples according to the L1 methylation levels, the global low methylation group showed poor prognosis despite high mutation load. In the combined lung cancer cohort ($n = 141$), the hazard ratio (HR) was 0.56 (log rank test, $P = 7.0 \times 10^{-3}$) (Fig. 5e). The IDIBELL cohort (n = 81) (Fig. 5f) and SMC cohort (left of

Fig. 5g) resulted in HR = 0.57 (log rank test, $P = 4.0 \times 10^{-2}$) and HR = 0.52 (log rank test, $P = 5.0 \times 10^{-2}$), respectively. Whereas the P values decreased as sample size grew, the effect size (HR) remained a similar level.

We next compared the effect of global methylation and mutation load on the clinical response. For the combined lung cancer cohort, methylome and matched exome data were available for 82 samples in total (Fig. 5a). In contrast to the global L1 methylation level (log rank test, HR = 0.46 and $P = 7.0 \times 10^{-3}$), mutation burden failed to show significant explanatory power (log rank test, HR = 0.77 and $P = 0.3$) (left and middle of Fig. 5h). Multiple regression with survival or clinical benefit as the response variable demonstrated significant effects by the global L1 methylation level but not mutation burden at $P = 0.05$ (right of Fig. 5h). We repeated the univariate

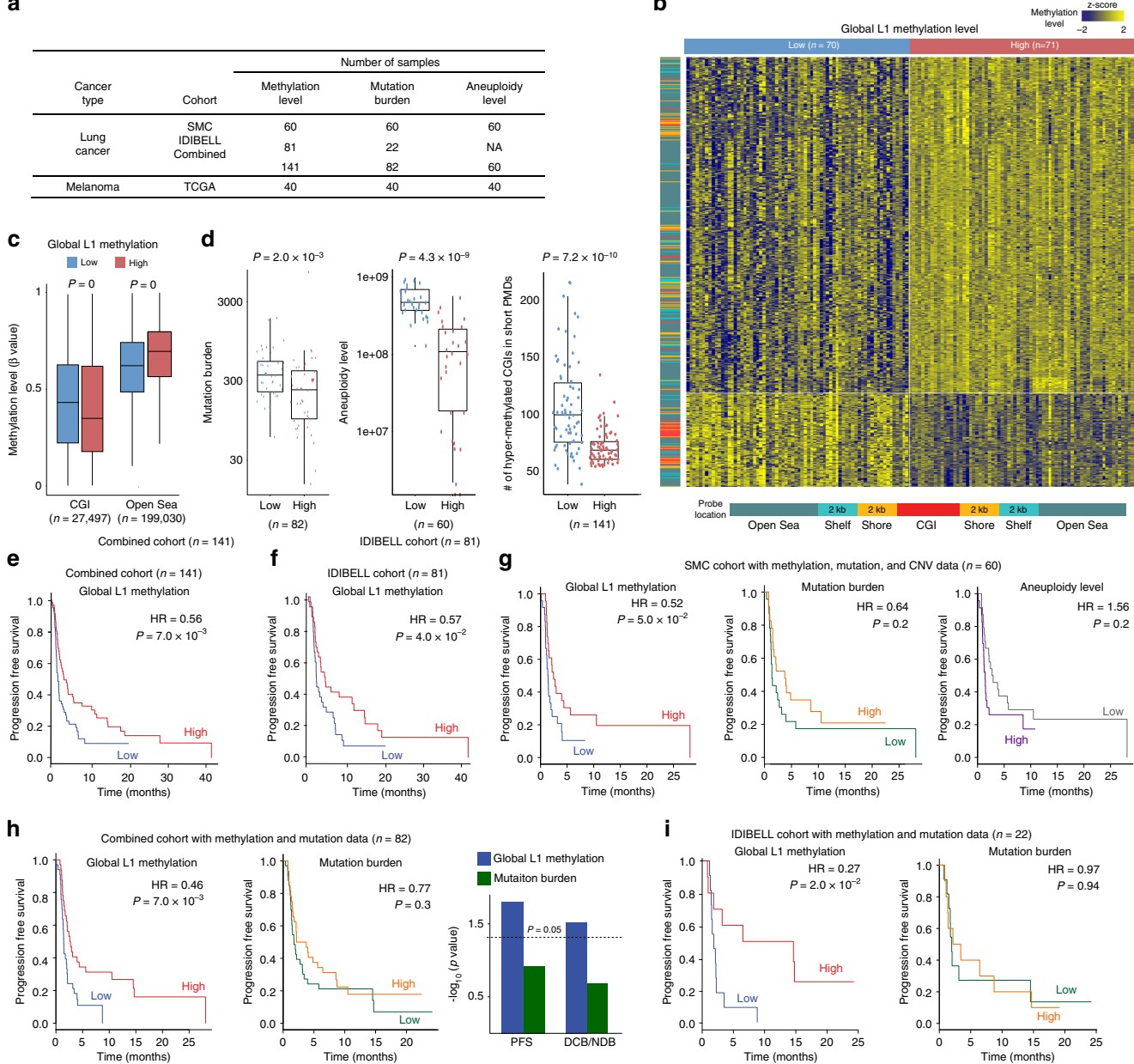

**Fig. 5** Genomic demethylation adversely affects the clinical benefit of checkpoint blockade. **a** Summary of immunotherapy cohort samples with available data. **b** Unsupervised hierarchical clustering of the DNA methylation profiles of the SMC lung cancer cohort samples. The methylation probes (row) were clustered and samples (column) were divided into two groups according to the median of the global methylation level. The heatmap shows beta values for the most differentially methylated loci (the highest 1%) between tumour samples with low and high global methylation. The methylation probes were categorized by the relative distance from CGIs (CGI, shore, shelf, and open sea). **c** Differential CGI/open-sea methylation between the global low and high groups. The differentially methylated loci between the two groups (FDR <0.05) were selected, and per-locus beta values were averaged for each group. The paired t-test was used to test the statistical significance. **d** Comparison of mutation burden, the aneuploidy level, and the number of hyper-methylated promoter CGIs in the short PMDs between tumour samples with low and high global methylation. **e**, **f** Survival analysis using tumour samples with methylation data from the **e** combined or **f** IDIBELL cohort. Patients were stratified by the global L1 methylation level. The log rank test was used to compare survival curves estimated by the Kaplan–Meier method. **g** Survival analysis using the SMC tumour samples with methylation level, mutation burden, and aneuploidy level. Patients were stratified by the global L1 methylation level (left), mutation burden (middle), and the aneuploidy level (right). **h** Survival analysis using tumour samples with methylation and mutation data from the combined cohort. Patients were stratified by the global L1 methylation level (left) or mutation burden (middle). To estimate the relative contribution of mutation burden and global methylation to patient survival and clinical benefit, the multivariate Cox proportional hazards model (for PFS: progression free survival) or multivariate logistic regression model (for DCB: durable clinical benefit and NDB: no durable benefit). **i** Survival analysis using the IDIBELL tumour samples with methylation and mutation data. Patients were stratified by the global L1 methylation level (left) or mutation burden (right)

comparisons for the two lung cancer cohorts separately. Both the SMC cohort ($n = 60$) and IDIBELL cohort ($n = 22$) resulted in significant stratification by global methylation but not by mutation burden (left and middle of Fig. 5g and i).

The global L1 methylation level negatively correlated with aneuploidy (middle of Fig. 5d). A pan-cancer analysis discovered the association of aneuploidy with signatures of immune evasion[10]. Taken together, both low methylation and high aneuploidy are expected to decrease tumour immunity and undermine the clinical benefit of immunotherapy. However, when we examined the SMC lung cancer cohort for which aneuploidy data were available (Fig. 5a), only global methylation but not aneuploidy showed significant correlations with poor clinical responses (left and right of Fig. 5g).

All the above analyses were repeated for the TCGA melanoma cohort ($n = 40$) (Fig. 5a). The samples were first divided into global low versus high methylation groups according to the L1 methylation levels. The global low methylation group in melanoma recapitulated CGI promoter hypermethylation in the short PMDs (Supplementary Fig. 7A) and poor prognosis in response to immunotherapies (log rank test, HR = 0.48 and $P = 3.0 \times 10^{-2}$) (Supplementary Fig. 7B). Mutation burden and aneuploidy level both failed to explain the clinical benefit (Supplementary Fig. 7C, D).

**Global demethylation rules out the effect of aneuploidy.** Our results on aneuploidy from the clinical data contradict the previous findings on the association of aneuploidy and immune evasion signatures[10]. Thus, we examined the possibility that global methylation is able to explain this association. Global demethylation in cancer promotes chromosomal instability[17–20]. DNA hypomethylation-related instability is mainly of chromosomal nature and involves large-scale alterations leading to aneuploidy rather than widespread amplifications or deletions[21–23]. As in ICF syndrome (for immunodeficiency, centromere instability, and facial anomalies), failure of methylation maintenance in pericentromeric sequences can cause erroneous chromosomal segregation in cancer[38–42].

Indeed, global demethylation significantly correlated with SCNAs across different tumour types (Fig. 6a). We determined the magnitude of chromosomal SCNAs (cSCNAs) by combining the chromosome SCNA and arm SCNA levels that were previously calculated[10], and compared this result with that of focal SCNAs (fSCNAs). The correlation was stronger with cSCNAs than with fSCNAs (Fig. 6b and Supplementary Fig. 8). We used partial correlations to estimate the extent to which the global L1 methylation level correlates with cSCNAs (or fSCNAs) when controlling for fSCNAs (or cSCNAs). In most cases, global hypomethylation was associated with cSCNAs independently of fSCNAs (Fig. 6c). In contrast, the correlation with fSCNAs disappeared when cSCNAs were controlled for (Fig. 6c). Importantly, the multiple regression analysis of the immune signature scores revealed markedly higher explanatory power for the global L1 methylation levels than cSCNA levels (Fig. 6d). We also performed the partial correlation analyses for the immune signature score, global L1 methylation level, and cSCNA level. Overall, the positive correlation between the immune signature score and global L1 methylation level maintained when the cSCNA level was controlled for (Fig. 6e), except for one tumour type that was previously reported as an exception regarding the role of aneuploidy[10]. These results indicate that the immune avoidance signatures of highly aneuploid tumours are associated with genomic demethylation. Indeed, a recent molecular mechanism study[43] contradicted the previous report[10] by suggesting that aneuploid cells generate pro-inflammatory signals

for their own elimination by the immune system as a means for cancer cell immunosurveillance.

## Discussion

In this work, we propose that DNA methylation aberration is an important determinant of the tumour response to host immune activity, and can provide a mechanism by which rapidly dividing and highly mutated tumours escape immune reaction and resist immunotherapy. The key mechanism seems to be the formation of heterochromatin, which is coupled with progressive domain-level methylation loss. An open question is what dictates these epigenetic changes in particular regions, such as the MHC locus. One possibility is that these changes at the particular loci are selectively favoured during cancer evolution because they provide immune evasion mechanisms and increase fitness of tumour cells. It is also possible that domain demethylation of immune-related regions is a more inherent chromatin tendency than a consequence of selection. In any case, our results suggest that mitotic cell division causes genetic and epigenetic alterations that exert opposing effects on tumour immunity by increasing neoantigens and inhibiting immune gene expression, respectively. Cell division also increases focal and chromosomal copy number changes. Our data show that the particular association of chromosomal copy number changes with low antitumour immune activity can be explained by global methylation loss.

There are multiple studies that reported antitumour immunity augmented by CDK4/6 inhibition and synergistic effects of the cell cycle inhibitors and checkpoint blockade[44–46]. Our results suggest that cell cycle inhibition may bring about opposing effects by suppressing genetic alterations that facilitate neoantigen formation and at the same time, preventing immune evasion promoted by epigenetic alterations that repress immunomodulatory pathway genes. Hence, the reported effects of cell cycle inhibition suggest that the benefits achieved by epigenetic influences may be greater than the adverse effects caused by suppressing neoantigen formation.

The repression signatures for IFN-α/β signaling draw particular attention, given that this pathway is supposed to stimulate immune responses against dsRNAs induced by genomic demethylation. Recent studies have shown that DNA methylation inhibitors induce dsRNA expression and stimulate antitumour immune activity through the IFN-α/β response activated by the viral defence pathway[27–29]. Based on these results, combining epigenetic therapy and immunotherapy has been suggested[47,48]. According to our results, tumours with global methylation loss tend to resist immunotherapy alone and may particularly benefit from this combined treatment approach. However, intrinsically de-suppressed dsRNAs may fail to boost antitumour immunity because of inactivated IFN-α/β signaling. We indeed observed that ERV/LINE expression does not increase the immune signatures, which, in contrast, are reduced probably reflecting IFN-α/β inactivation. Therefore, different action mechanisms of epigenetic therapy are required when targeting these tumours. Specifically, it needs to be tested whether methylation inhibitors or other epigenetic modulators are capable of restoring the IFN-α/β response and other immunomodulatory pathways by diminishing CGI methylation or loosening heterochromatin structure in these intrinsically demethylated tumours. Our study sheds light on the combination of epigenetic modulation and checkpoint blockade as a potential precision immunotherapy regimen.

## Methods

**TCGA molecular and clinical data.** The batch-corrected and normalized DNA methylation data based on Infinium Methylation 450k technology), together with mRNA expression and gene mutation data, generated by the PanCancer Atlas

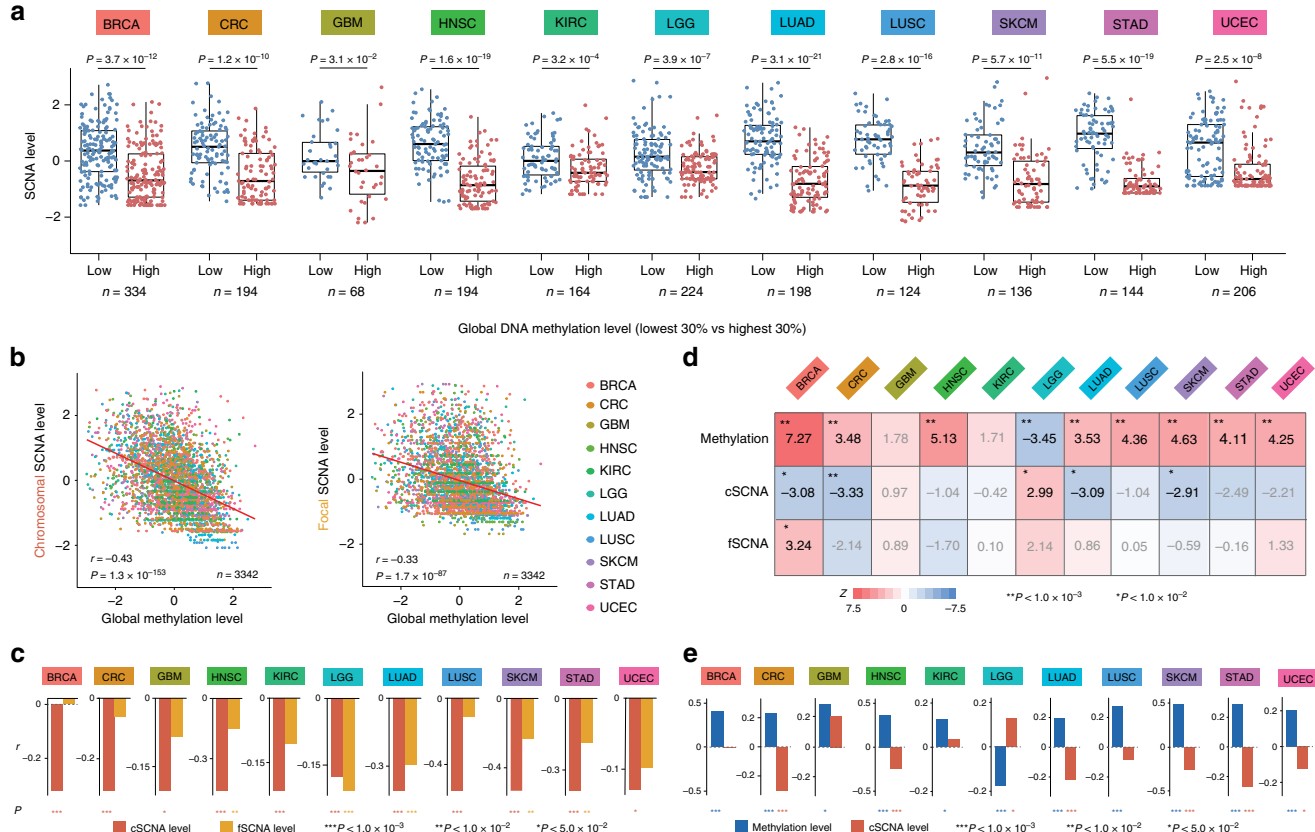

**Fig. 6** Aneuploidy indicates global DNA hypomethylation. **a** Comparison of SCNA levels between tumours with low and high global methylation. Shown here is the combination of cSCNAs and fSCNAs. *P* values from the two-sided Mann–Whitney U test are shown. **b** Association of global DNA methylation with cSCNA and fSCNA levels. The values were standardized per tumour type. Spearman's rank correlation coefficient and its *P* value are shown. **c** Partial correlation analyses comparing cSCNAs (red) and fSCNAs (orange) for their correlation with global methylation independently of one another. The Spearman correlation was used. **d** For each tumour type, samples with a low (<30th percentile) and high (>70th percentile) immune signature score were compared by multivariate logistic regression with the standardized global methylation, cSCNA, and fSCNA levels, and tumour purity as predictors. **e** Partial correlation analyses comparing global methylation (blue) and cSCNAs (red) for their correlation with the immune signature score independently of one another. The Spearman correlation was used. (**a**–**e**) These analyses were confined to 11 tumour types for which the cSCNA and fSCNA data were available

consortium were obtained from the publication page (https://gdc.cancer.gov/about-data/publications/pancanatlas). TCGA aliquot barcodes flagged as DO NOT USE in the Merged Sample Quality Annotation file were discarded. We selected cancer types for which there were >100 patient samples with all the molecular data and age information. The selected tumour types encompassed 6968 samples across 21 types (Supplementary Table 4), including bladder urothelial carcinoma (BLCA), breast adenocarcinoma (BRCA), cervical squamous cell carcinoma and endo-cervical adenocarcinoma (CESC), colorectal carcinoma (CRC), esophageal carci-noma (ESCA), head and neck squamous cell carcinoma (HNSC), kidney renal clear cell carcinoma (KIRC), kidney renal papillary cell carcinoma (KIRP), lower grade glioma (LGG), liver hepatocellular carcinoma (LIHC), lung adenocarcinoma (LUAD), lung squamous cell carcinoma (LUSC), pancreatic adenocarcinoma (PAAD), pheochromocytoma and paraganglioma (PCPG), prostate adenocarci-noma (PRAD), sarcoma (SARC), skin cutaneous melanoma (SKCM), stomach adenocarcinoma (STAD), testicular germ cell tumours (TGCT), thyroid carcinoma (THCA), and uterine corpus endometrial carcinoma (UCEC). PMD methylation levels were derived from Zhou et al.[16] (https://zwdzwd.github.io/pmd). Aneuploidy level and tumour purity values were obtained from a Table of Taylor et al.[49]. The cSCNA and fSCNA levels from 11 cancer types were obtained from Davoli et al.[10]. For glioblastoma multiforme (GBM), microarray-based gene expression data were obtained from the UCSC Xena public data hubs.

**Estimating global methylation levels**. To determine global methylation levels, we chose methylation probes for which at least 90% of sequences (≥45 bp) mapped to the young subfamilies of LINE-1 (L1HS and L1PA). Probe mapping information and LINE-1 family annotation were obtained from GPL16304[50] and the Repeat-Masker of the UCSC genome browser[51], respectively. We averaged the beta values of the chosen probes in each tumour sample, and used the average beta as an estimate of the global methylation level (Supplementary Data 1). For validation, we obtained the whole-genome bisulfite sequencing data (level 3 beta values) of samples for which the methylation array data were available (*n* = 18;

Supplementary Table 5) from the GDC legacy archive (https://portal.gdc.cancer.gov/legacy-archive). We first selected evolutionarily young LINE-1 repeat elements containing at least three different CpG sites that were covered by at least 10 aligned reads and then computed the averaged beta value for each repeat element. The number of the selected repeat elements for each sample ranged from 67,732 to 71,958 (Supplementary Table 5). We obtained the mean of the average beta values of the repeat elements to represent the global methylation level of each sample. We performed an additional validation by using the LINE-1 pyrosequencing dataset of 15 samples for which methylation array data of the same platform were available[52]. We compared our measures based on the selected array probes with the bisulfite sequencing and pyrosequencing measures (Supplementary Fig. 2).

**Linear regression modelling**. For each gene per cancer type, a linear regression model was fit using mRNA expression level as the response variable, and global methylation, mutation burden, aneuploidy, tumour purity, age, and tumour stage as predictors. We included tumour stage in the model for tumour types for which at least 100 patient samples with tumour stage information were available (*n* = 15; Supplementary Table 4). The regression model with the following formula was built using the lm function in R.

mRNA expression of gene Y ~ $\beta_1$ × global methylation level + $\beta_2$ × mutation burden + $\beta_3$ × aneuploidy level + $\beta_4$ × tumour purity + $\beta_5$ × age + $\beta_6$ × tumour stage

For each of three predictors (global methylation level, mutation burden, aneuploidy level) per tumour type, GSEA[48] was performed on genes with significant regression coefficients (Benjamini and Hochberg FDR <0.05). Genes ranked by the z score from the linear regression model were used for input into the preranked module of the GSEA software[48] with the immune and proliferation gene sets (see below). GSEA was run with default settings, except for the minimum number of gene sets, which was set to 10.

**Identification of genes with differential replication timing**. Repli-Seq mea-surements (wavelet-smoothed signal) of ENCODE 5 normal (HUVEC, IMR90,

NHEK, BJ_1 and BJ_2) and 10 cancer cell lines (MCF-7, SK-N-SH, HepG2, HeLa, A549, G401, LNCaP, T47D, H460, and Caki-2) were downloaded from the UCSC Genome Browser[53] and ENCODE project portal[54]. Average wavelet-smoothed signal values in each 5-kb window were scaled and quantile-normalized. Windows harbouring missing values in any of the cell lines were excluded. We performed the student's $t$-test for each 5-kb window to assess the replication timing difference between the normal and cancer cells. We then assigned the $P$ value and $t$ statistic to each gene. For genes spanning multiple replication timing windows, we assigned a combined $P$ value using the Fisher's method and average $t$ statistic. We excluded genes located on sex chromosomes or specifically expressed in the immune system (see below). Genes with Bonferroni-adjusted $P$ < 0.05 were defined as early- or late-replicated genes in cancer. Genes ranked by the $t$ statistic were used for input into the preranked module of the GSEA software[48] with the canonical REACTOME and KEGG pathways from MSigDB[55].

**Analysis of genes with differential replication timing.** To investigate the expression differences of early- and late-replicated genes between tumour samples with high and low global methylation levels, we partitioned tumour samples into the low (<30th percentile) and high (>70th percentile) global methylation groups for each cancer type. After a z-score normalization of mRNA expression data per gene per cancer type, we computed the average expression level of the genes for each group and compared the groups. To calculate the fraction of enriched cell-cycle and immune pathways (Fig. 3c), genes ranked by the $t$ statistic (replication timing difference between normal and cancer cells) were used for input into the preranked module of the GSEA software with REACTOME pathways (provided at https://reactome.org/) belonging to the Cell Cycle or Immune System category. Pathways harbouring at least 10 genes were used for this analysis.

**Clustering of PMD and analysis of focal hypermethylation in PMD.** By employing previously defined locations of PMDs and HMDs per 100 kb[16], we merged consecutive domains of the same type and retained those >300 kb in length. After assigning average methylation variability and replication timing from normal samples into each merged PMD, we performed hierarchical clustering on them. The enrichment of genes in particular pathways for clusters was computed using the binomial test. To estimate the number of hypermethylated CpG island promoter probes in short PMDs for each sample, we first calculated the mean and standard deviations of CpG island probes in the merged HMDs and then counted CpG island promoter probes (annotated as TSS200 or TSS1500) in short PMDs for which methylation level is greater than two standard deviations from the mean.

**Analysis of proximity of immune genes to PMD boundaries.** For a total of 77 immune genes in the pathways enriched for short PMDs ($n = 13$; Fig. 4c), we calculated the average distance to their nearest HMD (original HMD defined by Zhou et al.[16]) and then estimated its $P$ value by generating a background distribution. We randomly picked 77 PMD genes and calculated the average distance of them to the nearest HMD. This procedure was repeated 10,000 times.

**Collection and identification of marker gene sets.** We obtained markers for CD8 + T cells and proliferation (Fig. 1a, d) from Thorsson et al.[56] and used single-sample GSEA to estimate the activity of the markers. Gene sets for marking different types of immune cells and MHC (class I, class II, and non-classical) were derived from Charoentong et al.[57]. Hallmark immune and proliferation gene sets were obtained from MSigDB[55]. Antigen presentation and cytokine signaling pathways were derived from canonical KEGG pathway in MSigDB[55]. The gene set for the immune signature score was obtained from the aneuploidy study[10]. To filter out genes that are specifically expressed in the immune system, we used gene expression data from the Illumina's Human BodyMap 2.0 project (ftp://ftp.ncbi.nih.gov/gene/DATA/expression/Mammalia/Homo_sapiens/). Genes for which the average expression level in leukocytes and lymph nodes was five-fold higher than that in the remaining tissues ($n = 14$) were considered as genes specifically expressed in the immune system ($n = 1216$; Supplementary Data 4).

**Quantification of LINE-1 and ERV expression.** To quantify the LINE-1 and ERV expression levels, we aligned RNA-seq reads against the LINE-1[58] and ERV[59] sequence library, respectively, by using BWA[60]. We then normalized the mapped read counts by the total number of aligned RNA-seq reads. Reads that mapped to both of the libraries or other repeat libraries[58] (Alu and SVA) were excluded. The normalized expression levels were standardized per tumour type for comparison.

**Clinical data of the SMC cohort.** A total of 60 advanced non-small cell lung carcinoma patients who were treated with anti-PD-1/PD-L1 from 2014 to 2017 at Samsung Medical Center were enrolled for this study (Supplementary Table 3). The clinical response was evaluated by the Response Evaluation Criteria in Solid Tumours (RECIST) version 1.1 with a minimum 6-month follow-up. The response to immunotherapy was classified into durable clinical benefit (DCB, responder) or non-durable benefit (NDB, non-responder)[2]. Partial response (PR) or stable disease (SD) that lasted more than 6 months was considered as DCB/responder. Progressive disease (PD) or SD that lasted less than 6 months was considered as NDB/non-

responder. Progression-free survival (PFS) was calculated from the start date of therapy to the date of progression or death, whichever is earlier. Patients were censored at the date of the last follow-up for PFS if they were not progressed and alive. We complied with all relevant ethical regulations for work with human participants. Informed consent was obtained. This study was approved from the institutional review board at Samsung Medical Center (2018-03-130 and 2013-10-112).

**Whole-exome, transcriptome, and methylome data for the SMC cohort.** Tumour samples were obtained before anti-PD1/PD-L1 treatment, and then were embedded in paraffin after formalin fixation or kept fresh. DNA was prepared using AllPrep DNA/RNA Mini Kit (Qiagen, 80204), AllPrep DNA/RNA Micro Kit (Qiagen, 80284), or QIAamp DNA FFPE Tissue Kit (Qiagen, 56404) for library preparation for whole exome sequencing. Library preparation was performed by using SureSelectXT Human All Exon V5 (Agilent, 5190–6209) according to the instructions[61]. Briefly, 200–300 ng of tumour and normal genomic DNA was sheared, and 150–200 bp of the sheared DNA fragments were further processed for end-repairing, phosphorylation, and ligation to adaptors. Ligated DNA was hybridized using whole-exome baits from SureSelectXT Human All Exon V5. The libraries were quantified by Qubit and 2200 Tapestation, and sequenced on an Illumina HiSeq 2500 platform with $2 \times 100$ bp paired-ends. Target coverage for normal samples was $\times 50$ and tumour sample was $\times 100$.

The sequencing reads were aligned to the human reference genome (hg19) with BWA mem module (v0.7.12)[60] with default parameters. PCR duplicate reads were marked using Picard[62]. We used Strelka[63] to call somatic variants and selected single nucleotide variants (SNVs) and indels covered by at least ten and five reads in tumour, respectively. We further filtered out common germline variants present in dbSNP 150[64] and annotated somatic variants using ANNOVAR[65]. The list of filtered SNVs and indels is provided in Supplementary Data 5. Copy number variations (CNVs) were called using CNVkit[66] with the circular binary segmentation algorithm (Supplementary Data 6). Aneuploidy levels were derived from the called CNVs. Specifically, we applied the defined threshold of $\pm 0.2$ (average value of LUAD and LUSC) on the segment $\log_2$ ratio (tumour versus normal) to detect amplifications/deletions affecting at least 10% of a chromosome arm or 5% of a chromosome. The aneuploidy level was the sum of the absolute segment $\log_2$ ratio, each weighted by its length[10].

RNA was extracted from same tumour tissue using Allprep DNA/RNA Mini Kit (Qiagen, 80204). RNA was extracted from formalin fixed paraffin embedded (FFPE) using Rneasy FFPE kit (Qiagen, 73504). RNA was assessed for quality and quantity using nanodrop 8000 UV-Vis spectrometer (NanoDrop Technologies Inc) and 4200 TapeStation Instrument (Aglient Technologies). RNA integrity number (RIN) of > = 5 were selected for further library preparation. In total 500 ng of RNA from fresh tissues and 100 ng of RNA from FFPE were used for library preparation using Truseq RNA library prep kit v2 (Illumina, RS-122-2001, Rs-122-2002) or Truseq RNA access library prep kit (Illumina, RS-301-2001, RS-301-2002), respectively. The library was generated according to the manufacturer's instructions. RNA libraries were multiplexed and sequenced with 100 bp pair end reads on HiSeq2500 platform (Illumina).

The RNA-seq reads were aligned to the human reference genome (hg19) with STAR[67] and gene expression values were quantified using RSEM[68]. Genes ranked by $t$-values obtained from comparing mRNA expression levels between tumours with low ($n = 14$) and high global methylation level ($n = 13$) were used for input into the preranked module of the GSEA software with KEGG pathways and the MHC gene set.

Methylation assay was performed by following the instructions of Infinium MethylationEPIC BeadChIP Kit (Illumina, WG-317-1002). Briefly, 500 ng genomic DNA (gDNA) was used for bisulfite conversion using the EZ DNA methylation kit (Zymo Research, D5001). The bisulfited gDNA was denatured and neutralized for amplification, and was further processed for fragmentation. After fragmentation, DNA was eluted and resuspended in a hybridization buffer, and then hybridized onto the BeadChip. The BeadChip was prepared for staining and extension after washing out unhybridized DNA, and it was imaged using the Illumina iScan System. The raw intensity files were then preprocessed into beta values using the preprocessIllumina function in minfi[69]. The methylation data were treated as described in the Estimating global methylation levels section. The PMD levels of our cohort samples were calculated based on the average of EPIC probes for Solo-WCGW CpGs in common PMDs[16] (provided at https://zwdzwd.github.io/pmd). Redundant probes such as multi-hit probes by using the filter function of the ChAMP package[70]. We used MethylCIBERSORT[71] and ESTIMATE[72] to estimate tumour purity and leukocyte fraction (Supplementary Fig. 4). We processed the raw methylation intensity files of 81 lung cancer samples of the IDIBELL cohort[37] with the same pipeline and merged them with the SMC cohort data using ComBat[73].

**Melanoma cohort data.** Progression-free survival data for melanoma patients who received immune checkpoint inhibitors (drug name labelled as Ipilimumab, Yervoy, or Pembrolizumab; $n = 15$) were obtained from Ock et al.[9]. We included additional 25 patients that received other types of immunotherapy using drug data from the GDC legacy archive. We selected samples for which the therapy type (CDE_ID:2793530) column indicated immunotherapy while excluding samples from patients that received multiple drugs. The molecular data for these samples were obtained as described in the TCGA molecular and clinical data section.

**Multivariate survival analysis**. Global methylation and mutation burden were combined in a multivariable Cox proportional hazards model using the coxph function in R. The multivariable logistic regression model was used to assess the impact of global methylation and mutation burden on the objective response using the glm function in R.

**Reporting summary**. Further information on research design is available in the Nature Research Reporting Summary linked to this article.

## Data availability
The methylation chip and RNA-seq data for the samples of our lung cancer cohort are available at Gene Expression Omnibus under GSE119144 and GSE135222, respectively. The raw data for the exome sequencing of our SMC cohort samples have been submitted to European Genome-phenome Archive under accession number EGAS00001003731.

## Code availability
Computer codes used in this study are provided as Supplementary Software 1.

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

## Acknowledgements

This work was supported by the Post-Genome Technology Development Program [10067758, Business model development driven by clinico-genomic database for precision immuno-oncology] funded by the Ministry of Trade, Industry and Energy (MOTIE, Korea). H.J. was supported by a grant from the National Research Foundation of Korea funded by the Korean Government (NRF-2018R1A6A3A01010889).

## Author contributions

H.J. performed all data analyses and wrote the manuscript. H.S.K. generated all data and helped with manuscript writing. J.Y.K. participated in data analyses. J.-M.S., J.S.A., M.-J.A., K.P. and M.E. helped with cohort data generation and analyses. S.-H.L. and J.K.C. conceived and supervised the study.

## Additional information

**Competing interests:** The authors declare no competing interests.

