## [Peer Review File · Nature Communications]

Reviewers' Comments:

Reviewer #1:

Remarks to the Author:

Background and significance: There is a great deal of interest in what mechanisms lead to immune evasion, and what related biomarkers may be useful in predicting effective immune activation for checkpoint blockade therapies. Single-nucleotide mutation density, or tumor mutation burden (TMB), is only weakly predictive of response, and it was recently shown that a much stronger predictor is the frequency of chromosome or arm-level somatic copy number variants (sCNAs), or aneuploidy (Davoli et al 2017, PMID 28104840). Aneuploidy was shown to be negatively correlated with expression of immune signature genes and immune cell infiltration, and predicted poorer response to checkpoint blockade therapy. The relative contributions are somewhat difficult to disambiguate, since both are associated with proliferation.

A third property linked to tumor proliferation is DNA methylation, which was not investigated in earlier studies. Global hypomethylation occurs in tumors and has been shown to be correlated with mutation density and cell proliferation markers. Furthermore, most demethylation occurs within "Partially Methylated Domains" (PMDs) associated with late replicating heterochromatin. This study investigates the association between global demethylation and immunity pathways using expression data in TCGA, and response to checkpoint blockade therapy within a cohort of 40 lung cancer cases. They show that hypomethylation is associated with silencing of immune pathways, and that key immunity genes occur within hypomethylated PMDs. They show that while global hypomethylation is correlated with aneuploidy, methylation appears to be more strongly predictive of response to therapy. If proven out, this would be a major clue into how tumors evade immune surveillance, and could have implications for precision therapy.

Overall assessment: While this is potentially an important finding, I have two major concerns which would need to be addressed. First and most importantly, the sample size of the immune blockade cohort is too small to disentangle the 3 highly correlated markers (TMB, aneuploidy/sCNA, and methylation). The correlative evidence here trends in the right direction, but is not strongly convincing on its own. Second, it has been shown (and the authors reaffirm) that global hypomethylation is correlated with focal hypermethylation of CpG island promoters. In my opinion, CpG island hypermethylation is just as or more likely to be the source of immune pathway silencing than global hypomethylation. Yet this feature is not explored in detail, only global hypomethylation.

Major comments:

Overall, I find the sample numbers for immune checkpoint inhibitor cohorts to be too small to disentangle the contribution of the highly correlated features. For instance, methylation alone seemed to have some effect in the normal vs. aberrant methylome analysis (Figure 5D), but global methylation did not show much observable effect in the relative contributions model (Figure 6D). Likewise, you did not seem to have power to detect significant effects in overall survival in the 3-feature multivariate Cox model (Supp Figure 4F). Survival analysis of Aneuploidy alone is not shown, and can thus not be compared to the larger studies of aneuploidy (Davoli, 2017). Likewise, survival analysis for global methylation alone is not shown.

The melanoma cohort was even smaller (15 cases), and is not big enough to be helpful in my opinion. Compare the sample numbers here to those used to establish the link with aneuploidy (Davoli, 2017) - which used two independent melanoma cohorts, one with n=110 and one with n=64. Disentangling a third linked feature would presumably require even larger samples sizes.

Taylor et al. 2018 (PMID 29622463) showed that the correlation between sCNA and immune gene expression went away when adding in the leukocyte fraction to the regression model. You have tried to control for this by removing genes expressed in immune cells, but you should also explicitly check that your association with global methylation is not dependent on leukocyte

fraction.

You seem to propose a mechanism where global demethylation may be responsible for the silencing of immune response genes. For instance, on p.5, line 130, you say that "global demethylation may transcriptionally repress this response". On p.5, line 139, you say that "methylation loss in late-replicating regions engages the formation of heterochromatic structure". On p.10, line 259 you say that they "key mechanism seems to be the formation of heterochromatin". Yet this is not my understanding of what's known about PMD hypomethylation. Most data suggest that these regions are already organized as heterochromatin (for instance late replicating, associated with nuclear lamina, and silencing marks such as H3K9me3) in normal cells, and that DNA is hypomethylated as a consequence. A recent paper, Zhou et al 2018 (PMID 29610480) suggests that this may be a passive result of their late replication. A subset of genes within these domains, those with poised/polycomb bound CpG island promoters, do get hypermethylated and downregulated in an "epigenetic switch" from polycomb to DNA methylation (Gal-Yam et al. 2008, PMID 18753622). But aside from those specific genes, I do not know of any evidence that demethylation of PMDs leads to downregulation. I think your study would benefit greatly from characterizing the methylation levels of poised/polycomb CpG island promoters as a distinct biofeature. Are many / most of your downregulated immunity genes associated with this class?

Related, p. 6, lines 155-157, " Indeed, the late-replicating genes repressed in demethylated tumours (Fig. 3A) were enriched in the PMDs or common PMDs in contrast to HMDs 15 (Fig. 3F). The INF- α family genes were also in the PMDs or common PMDs (Fig. 3E)." Do you mean to say that these genes have CpG Island promoters which are hypermethylated in globally demethylated tumors? That is what I infer based on the preceding sentence. If so, you should show this explicitly.

A mechanistic link between aneuploidy and immunosensing was shown in Santaguida 2018 (Dev Cell)? How do you account for that?

Other comments:

In the abstract you state, "Here we find that markers of cell division correlate also with genomic demethylation involving methylation loss in late-replicating partial methylation domains". This implies that this is a new finding, but I believe that was one of the conclusions of Zhou et al 2018 (PMID 29610480). They correlated global demethylation to expression of cell proliferation markers and tumor mutation density.

P.5, line 131. " Without the silencing of the IFN- α/β pathway, genomic methylation would cause the antiviral response". Did you mean to write "genomic demethylation" ?

p.6, line 154, "supporting the previous reports on PMD gene silencing". Which reports do you refer to here?

p. 31 line 856. References #56 and #57 are identical.

In Figure 5-6 and Supp Fig 5, some of the survival plots show HR and some do not. For instance, Figure 5d and Supp Fig 5A do not show HR.

Reviewer #2:

Remarks to the Author:

The authors set out to explore an important question- namely the impact DNA hypomethylation on

tumor immune evasion. To do this they begin by evaluating correlations in TCGA data between proliferation, mutation, aneuploidy and CD8+ T cells with methylation level for different tumor types and show that hypomethylation is correlated with proliferation, mutation rate and SCNAs. To further evaluate the connection between hypomethylation and immune evasion, they perform a regression of gene expression for sample-level features in an attempt to prove that immune infiltrates and pathways associate with global methylation levels. They also evaluate expression of genes in early and late replicating portions of the genome (of which the latter is reported to be hypomethylated), drawing from cell line cell cycle data) and show hypomethylation is associated with transcriptional repression. The authors go on to report that immune genes are overrepresented in PMDs and that hypomethylation affects response to checkpoint blockade and has a predictive power. The authors' connection between hypomethylation and immune evasion is intriguing, but many of their points are weakened because they have been previously described and/or they are merely correlations.

Major points:

-The most significant criticism of this paper is that functional validation of the proposed mechanism of PMD-dependent immune gene silencing has not been performed. In addition, the authors have not demonstrated that the proposed mechanism of immune evasion (silencing of the HLA locus primarily) is operative in the immunotherapy cohorts they include - are HLA genes silenced in hypomethylated and checkpoint-unresponsive tumors?

-The first two figures report on correlations alone- many of which have been described (connection between SCNA and methylation). Also, the authors assume an average methylation level for the tumor types- ignoring that many tumors show varying degrees of hypomethylation. It would be good to consider variability of methylation levels within tumors. -The analyses presented in figures 1a-1d are interesting. However, there are numerous confounders between different tumor types that would be difficult to control for. Do these correlations hold up within single tumor types? For example, if melanoma samples with low vs high global methylation levels are compared, is there still a correlation with CD8 infiltration, etc.?

-The authors state "to focus on the repression of immune-response genes in tumour cells, we excluded genes that are specifically expressed in the immune system from the following analyses." To argue for a tumor cell intrinsic mechanism from bulk data that connects hypomethylation with repression, looking at genes specifically expressed in the tumor would be preferable.

-Figure 3 looks at early and late replicating regions based on cell line data and then explores expression of genes in these regions, extrapolating that since PMDs/tumor hypomethylation are reported to be correlated, expression of genes in these late replicating regions would reflect hypomethylation. A better strategy would be to look at expression levels in regions that are hypomethylated, rather than use a different measure (replication timing) in a different system (cell lines) to evaluate expression trends due to hypomethylation.

-The authors argue that their finding of repression in late replicating regions of demethylated tumors is in contrast to studies showing de-methylating agents upregulate IFNalpha/beta response genes- but fail to consider the effect these agents have on island hypermethylation (which is a putative mechanism for upregulation of genes in this experimental setting).

-Figure 4 looks at PMD methylation as a proxy of overall hypomethylation- this is consistent with previous data that have shown PMDs undergo hypomethylation in tumors; the novelty of this point is weak.

-An intriguing finding in this paper is that the survival upon immune checkpoint blockade differs between more and less hypomethylated tumors. The authors should expand their analysis more to

evaluate this in other tumors given the relatively small sample size in their cohort. It would also be interesting to evaluate this in other tumor types. Finally, they should control for stage, age, genotype, etc. Do the tumors with longer survival downregulate HLA genes?

-Figure 7 looks at correlations between SCNA and hypomethylation- which has previously been described. They argue their results show that immune avoidance signatures of highly aneuploid tumors is explained by demethylation- but the language needs to be precise here. It is associated with demethylation but on their analysis alone, it is not possible to argue that it is explained by this (ie. Causation is not proven).

Reviewer #3:

Remarks to the Author:

In this manuscript, Jung et al., investigate the relationship between global DNA methylation levels and immunological parameters as well as response to immunotherapy using TCGA data and a small lung cancer data-set (n=40). They conclude that global DNA methylation levels is a predictive marker in immunotherapy and imply that epigenetic therapy or cell cycle inhibition could be used in combination with immunotherapy.

I have several major concerns regarding this manuscript.

1 – It needs some revision to increase clarity. The use of a professional English editor could help improve clarity and readability.

2 - The global methylation score was based on the array probes that map to LINE1 elements. They validated this score using WGBS for 18 samples that have WGBS and array data available. However, for the WGBS, it was used only CpGs that map to LINE1 elements. Therefore, it was not validated the global methylation levels, only validated the global LINE1 methylation level. It would make more sense to compare the methylation score against all the CpGs in the WGBS data. Alternatively, the authors should not call it 'global methylation level' but rather 'global LINE1 methylation levels'.

3 – Little data is provided on how all the methylation data was normalized from TCGA. Each TCGA cancer type was profiled separately and batch correction is necessary for PAN cancer analysis.

4 – The global methylation level will be modified by the amount of tumor purity and amount of different immune cells per tumors. Did the authors correct for purity and for the different immune infiltrates? I suggest to use ABSOLUTE to calculate purity and MethyCYBERSORT to calculate the immune-infiltrate from the methylation data. Then correct for this when do the global analysis.

5 – For instance, the correlation between DNA methylation levels and tumour-infiltrating CD8+ T cells (Fig 1D), could suggest that global tumor hypomethylation is correlated with CD8 infiltration (authors conclusion) or, alternatively, that CD8 T cells have global hypomethylated DNA, and the more CD8 T cells present in the biopsied sample, more demethylation will be observed in the bulk data. This needs to be addressed.

6 – For the Lung cohort, how the 'normal' and 'aberrant' methylation samples were selected? What probes were used? How normal and aberrant was defined? How each sample was assigned to each group?

7- I disagree with the authors conclusions about cell cycle inhibitors in the discussion. These

patients already have the DNA methylation change before treatment. Cell cycle inhibitors will not prevent or revert these changes, even if the demethylation was initially caused by cell divisions.

Minor:

8 – Reference #28 and #44 are the same.

Reviewers' comments:

Reviewer #1 (Remarks to the Author):

Background and significance: There is a great deal of interest in what mechanisms lead to immune evasion, and what related biomarkers may be useful in predicting effective immune activation for checkpoint blockade therapies. Single-nucleotide mutation density, or tumour mutation burden (TMB), is only weakly predictive of response, and it was recently shown that a much stronger predictor is the frequency of chromosome or arm-level somatic copy number variants (sCNAs), or aneuploidy (Davoli et al 2017, PMID 28104840). Aneuploidy was shown to be negatively correlated with expression of immune signature genes and immune cell infiltration, and predicted poorer response to checkpoint blockade therapy. The relative contributions are somewhat difficult to disentangle, since both are associated with proliferation.

A third property linked to tumour proliferation is DNA methylation, which was not investigated in earlier studies. Global hypomethylation occurs in tumours and has been shown to be correlated with mutation density and cell proliferation markers. Furthermore, most demethylation occurs within "Partially Methylated Domains" (PMDs) associated with late replicating heterochromatin. This study investigates the association between global demethylation and immunity pathways using expression data in TCGA, and response to checkpoint blockade therapy within a cohort of 40 lung cancer cases. They show that hypomethylation is associated with silencing of immune pathways, and that key immunity genes occur within hypomethylated PMDs. They show that while global hypomethylation is correlated with aneuploidy, methylation appears to be more strongly predictive of response to therapy. If proven out, this would be a major clue into how tumours evade immune surveillance, and could have implications for precision therapy.

Overall assessment: While this is potentially an important finding, I have two major concerns which would need to be addressed. First and most importantly, the sample size of the immune blockade cohort is too small to disentangle the 3 highly correlated markers (TMB, aneuploidy/sCNA, and methylation). The correlative evidence here trends in the right direction, but is not strongly convincing on its own. Second, it has been shown (and the authors reaffirm) that global hypomethylation is correlated with focal hypermethylation of CpG island promoters. In my opinion, CpG island hypermethylation is just as or more likely to be the source of immune pathway silencing than global hypomethylation. Yet this feature is not explored in detail, only global hypomethylation.

Major comments:

1. Overall, I find the sample numbers for immune checkpoint inhibitor cohorts to be too small to disentangle the contribution of the highly correlated features. For instance, methylation alone seemed to have some effect in the normal vs. aberrant methylome analysis (Figure 5D), but global methylation did not show much observable effect in the relative contributions model (Figure 6D). Likewise, you did not seem to have power to detect significant effects in overall survival in the 3-feature multivariate Cox model (Supp Figure 4F). Survival analysis of aneuploidy alone is not shown, and can thus not be compared to the larger studies of aneuploidy (Davoli, 2017). Likewise, survival analysis for global methylation alone is not shown. The melanoma cohort was even smaller (15 cases), and is not big enough to be helpful in my opinion. Compare the sample numbers here to those used to establish the link with aneuploidy (Davoli, 2017) - which used two independent melanoma cohorts, one with n=110 and one with n=64. Disentangling a third linked feature would presumably require even larger samples sizes.

We agree with the reviewer that our initial sample size was not large enough for multivariate analyses. As pointed out by the reviewer, the relative effect of DNA methylation was not consistently higher than that of mutation burden (previous Figure 6C and 6D) in contrast to the result of the univariate analyses (previous Figure 5D vs Figure 6B). A similar pattern is observed in the aneuploidy study (Davoli, 2017) as attached below: the univariate analyses show that only the SCNA level correlates with patient survival (the survival plots below); however, the multivariate survival analysis suggests a more significant contribution of mutation burden (the table below).

SURVIVAL ANALYSIS: MUTATIONS AND LEVEL OF SCNAs

Parameter	HR	Wald test p-value
N of mutations	0.497	0.00039
Level of SCNAs	3.258	0.0031
Combined Score	3.61(1.65,7.87)	0.0012

(Davoli et al. PMID 28104840)

We made considerable efforts to increase sample size. By following the workflow shown below, we were able to collect 60 high-quality DNA samples that passed quality control for both methylome and exome analyses from non-small cell lung carcinoma (NSCLC) patients who were

treated with anti-PD-1/PD-L1. We named our cohort after Samsung Medical Center (SMC).

However, as seen in the above example (n = 64 from Davoli, 2017), the sample size of 60 still deemed not sufficient. We thus looked to include published data for validation and meta-analysis. Our thorough literature search led us to an anti-PD-1 NSCLC cohort composed of 81 methylomes and 22 mutanomes (Duruiseaux et al. PMID:30100403) from IDIBELL (Bellvitge Biomedical Research Institute). In addition, we increased the size of the TCGA melanoma cohort to 40 by including other types of immunotherapies. The following table is the summary of the resulting cohorts.

Cancer type	Cohort	Number of samples		
		Methylation level	Mutation burden	Aneuploidy level
Lung cancer	SMC	60	60	60
	IDIBELL	81	22	NA
	Combined	141	82	60
Melanoma	TCGA	40	40	40

The following are univariate analysis results for the SMC cohort:

The following are univariate analysis results for the IDIBELL cohort:

It is notable that the two independent data agree on the importance of global methylation, but not mutation burden.

The following are univariate analysis results for the combined cohort:

Because the sample size of the combined cohort was > 80 (n = 82), we attempted a multivariate analysis for methylation and mutation burden. As shown below, only global methylation, but not mutation burden, was a significant contributor to patient survival (left) and clinical benefit (right).

The melanoma cohort with increased sample size produced the results consistent with the previous cohort that contained checkpoint blockade samples only.

It should be also emphasized that the results of molecular-level analyses shown in Figure 2 were derived from the following multivariate regression across thousands of pan-cancer samples.

mRNA expression of gene Y $\sim \beta_1$ * global methylation level + β_2 * mutation burden + β_3 * aneuploidy level + β_4 * tumour purity + β_5 * age + β_6 * tumour stage

In this manner, we were able to determine that immune infiltrates are associated with global methylation levels independently of mutation burden and aneuploidy when purity, age, and tumour stage are adjusted (Figure 2A). Significant correlations with genomic demethylation were observed also for immunomodulatory pathways that should include genes expressed in tumour cells, independently of mutation burden and aneuploidy (Figure 2B). These include antigen processing and presentation, MHC, cytokine-cytokine receptor interaction, interferon or other cytokine signaling, and complement and coagulation.

2. Taylor et al. 2018 (PMID 29622463) showed that the correlation between SCNA and immune gene expression went away when adding in the leukocyte fraction to the regression model. You have tried to control for this by removing genes expressed in immune cells, but you should also explicitly check that your association with global methylation is not dependent on leukocyte fraction.

Following the reviewer's suggestion, we checked whether the correlation between global methylation level and immune gene expression would go away when adding the leukocyte

fraction to the regression model. The leukocyte fraction was calculated based on gene expression and methylation data using ESTIMATE (Yoshihara et al. PMID:24113773) and MethylCIBERSORT (Chakravarthy, PMID: 30104673), respectively. As seen below, whereas the addition of the leukocyte fraction resulted in the loss of enrichment for immune genes, the correlation between global methylation level and proliferation gene expression maintained. These results confirm that the global methylation level itself is not affected by the leukocyte fraction, which correlates only with immune gene expression as expected. These results have been added to Supplementary Fig. 6.

3. You seem to propose a mechanism where global demethylation may be responsible for the silencing of immune response genes. For instance, on p.5, line 130, you say that "global demethylation may transcriptionally repress this response". On p.5, line 139, you say that "methylation loss in late-replicating regions engages the formation of heterochromatic structure". On p.10, line 259 you say that they "key mechanism seems to be the formation of heterochromatin". Yet this is not my understanding of what's known about PMD hypomethylation. Most data suggest that these regions are already organized as heterochromatin (for instance late replicating, associated with nuclear lamina, and silencing marks such as H3K9me3) in normal cells, and that DNA is hypomethylated as a consequence. A recent paper, Zhou et al 2018 (PMID 29610480) suggests that this may be a passive result of their late replication. A subset of genes within these domains, those with poised/polycomb

bound CpG island promoters, do get hypermethylated and downregulated in an "epigenetic switch" from polycomb to DNA methylation (Gal-Yam et al. 2008, PMID 18753622). But aside from those specific genes, I do not know of any evidence that demethylation of PMDs leads to downregulation. I think your study would benefit greatly from characterizing the methylation levels of poised/polycomb CpG island promoters as a distinct biofeature. Are many / most of your downregulated immunity genes associated with this class?

Because the reviewer suggests that most PMDs are already in heterochromatin states in normal cells, we sought to test whether there are different types of PMDs in normal cells. In doing so, we identified three distinct subclasses of PMDs in normal cells based on methylation variability and replication timing. We used methylation variability because PMDs were characterized by and defined based on the high variability of solo-WCGW methylation levels across samples (Zhou et al. PMID:29610480). Among the three subclasses, immune genes were significantly enriched in the class of weak PMDs that were characterized by a relatively lower variability, earlier replication timing, and smaller domain size than the other two subclasses.

The weak PMD class seems to coincide with the previously identified PMDs that replicate during the early and middle S phases (Salhab et al. 2018, PMID : 30266094). In light of this

finding, we conclude that genes in those PMDs that are similar to HMDs in normal cells might become silenced as a consequence of malignancy.

To explore the ‘epigenetic switch’ mechanism, we selected CpG island methylation probes on the bivalent promoters that were annotated by at least 10 different samples in the Roadmap Epigenomic Project. We found a significant over-representation of these bivalent CpG island promoters in the weak PMDs where immune genes were concentrated ($P = 4.0 \times 10^{-247}$ from the hypergeometric test). We then sought to test whether the bivalent promoters are hypermethylated in globally demethylated tumours. To this end, we compared the numbers of hypermethylated probes between tumours with low and high global methylation levels. Whereas globally demethylated tumours carried a significantly larger number of hypermethylated CpG island promoters, the discrepancies were more prominent for the bivalent promoters (see the graph below). This finding supports the ‘epigenetic switch’ mechanism underlying CpG island promoter hypermethylation in PMDs.

4. Related, p. 6, lines 155-157, "Indeed, the late-replicating genes repressed in demethylated tumours (Fig. 3A) were enriched in the PMDs or common PMDs in contrast to HMDs (Fig. 3F). The INF- α family genes were also in the PMDs or common PMDs (Fig. 3E)." Do you mean to say that these genes have CpG Island promoters which are hypermethylated in globally demethylated tumours? That is what I infer based on the preceding sentence. If so, you should show this explicitly.

In response to this constructive comment, we explicitly show that the PMD genes have CGI promoters that are hypermethylated and thus are repressed in globally demethylated tumours in our revised Fig. 3A, Fig. 3B, Fig. 4F, and Fig. 4G. In the weak PMDs (Fig. 4F and Fig. 4G) and late-replicating regions (Fig. 3A and Fig. 3B), the number of hypermethylated CpG Island promoters were significantly higher and the expression levels were lower in tumours with low global methylation levels than in tumours with high global methylation levels. We attach those results below.

<Hypermethylation of CGI promoters in the weak PMDs in demethylated tumours>

<Repression of genes in the weak PMDs in demethylated tumours>

<Hypermethylation of CGI promoters in late-replicating regions in demethylated tumours>

<Repression of genes in late-replicating regions in demethylated tumours>

These results are consistent with multiple previous reports (listed up below) showing that genes in PMDs in cancers are largely silenced by the formation of repressive chromatin structures or via CGI hypermethylation.

Berman, B. P. *et al.* Regions of focal DNA hypermethylation and long-range hypomethylation in colorectal cancer coincide with nuclear lamina-associated domains. *Nat. Genet.* **44**, 40–46 (2012).

Hon, G. C. *et al.* Global DNA hypomethylation coupled to repressive chromatin domain formation and gene silencing in breast cancer. *Genome Res.* **22**, 246–58 (2012).

Hovestadt, V. *et al.* Decoding the regulatory landscape of medulloblastoma using DNA methylation sequencing. *Nature* **510**, 537–541 (2014).

Timp, W. *et al.* Large hypomethylated blocks as a universal defining epigenetic alteration in human solid tumors. *Genome Med.* **6**, 61 (2014).

Brinkman, A. B. *et al.* Partially methylated domains are hypervariable in breast cancer and fuel widespread CpG island hypermethylation. *Nat. Commun.* **10**, 1749 (2019).

5. A mechanistic link between aneuploidy and immunosensing was shown in Santaquida 2018 (Dev Cell)? How do you account for that?

We thank the reviewer for referring us to this important study. We find that the results suggested by Santanuida et al. (PMID:28633018) are actually in conflict with those by Davoli et al. (PMID 28104840). Santanuida et al. report the upregulation of genes that mediate inflammation and an immune response in aneuploid cells. What they show is that cells with abnormal karyotypes generate a signal for their own elimination that may serve as a means for cancer cell immunosurveillance. If this is true, highly aneuploid tumours should show good prognosis in checkpoint blockade immunotherapy, which is contradicting what Davoli et al. suggest. What we can conclude by combining all results together is that aneuploid itself seems to increase tumour immunity (Santanuida et al.), but that its coupling with global methylation loss explains the immune evasion of aneuploid tumours (Davoli et al.). We add some statements regarding this point to the Discussion as follows.

(Page 12, Line 10) These results indicate that the immune avoidance signatures of highly aneuploid tumours ~~can be explained by~~ are associated with genomic demethylation. ~~Indeed, a recent molecular mechanism study⁴¹ contradicted the previous report¹⁰ by suggesting that aneuploid cells generate pro-inflammatory signals for their own elimination by the immune system as a means for cancer cell immunosurveillance.~~

Other comments:

6. In the abstract you state, "Here we find that markers of cell division correlate also with genomic demethylation involving methylation loss in late-replicating partial methylation domains". This implies that this is a new finding, but I believe that was one of the conclusions of Zhou et al 2018 (PMID 29610480). They correlated global demethylation to expression of cell proliferation markers and tumour mutation density.

We modified the abstract as follows:

Mitotic cell division increases tumour mutation burden and copy number load with a positive and inverse correlation, respectively, with the clinical benefit of immunotherapy. ~~Here we find that~~ Markers of cell division correlate also with genomic demethylation involving methylation loss in late-replicating partial methylation domains. ~~Here we find that~~ immunomodulatory pathway genes are concentrated in these domains and transcriptionally repressed in demethylated tumours with CpG island hypermethylation.....

7. P.5 , line 131. " Without the silencing of the IFN- α/β pathway, genomic methylation would cause the antiviral response". Did you mean to write "genomic demethylation" ?

We thank the reviewer for pointing this out. It has been corrected to genomic demethylation.

(Page 6, Line 19) Without the silencing of the IFN- α/β pathway, genomic demethylation would cause the antiviral response and facilitate antitumour immune reaction as demonstrated with demethylating agents.

8. p.6, line 154. "supporting the previous reports on PMD gene silencing". Which reports do you refer to here?

We meant Berman et al. 2012. PMID: 22120008. This statement has been deleted in our revised manuscript.

9. p. 31 line 856. References #56 and #57 are identical.

Corrected.

10. In Figure 5-6 and Supp Fig 5, some of the survival plots show HR and some do not. For instance, Figure 5d and Supp Fig 5A do not show HR.

HR is now shown in all revised plots.

Reviewer #2 (Remarks to the Author):

The authors set out to explore an important question- namely the impact DNA hypomethylation on tumour immune evasion. To do this they begin by evaluating correlations in TCGA data between proliferation, mutation, aneuploidy and CD8+ T cells with methylation level for different tumour types and show that hypomethylation is correlated with proliferation, mutation rate and SCNAs. To further evaluate the connection between hypomethylation and immune evasion, they perform a regression of gene expression for sample-level features in an attempt to prove that immune infiltrates and pathways associate with global methylation levels. They also evaluate expression of genes in early and late replicating portions of the genome (of which the latter is reported to be hypomethylated), drawing from cell line cell cycle data) and show hypomethylation is associated with transcriptional repression. The authors go on to report that immune genes are overrepresented in PMDs and that hypomethylation affects response to checkpoint blockade and has a predictive power. The authors' connection between hypomethylation and immune evasion is intriguing, but many of their points are weakened because they have been previously described and/or they are merely correlations.

Major points:

1. The most significant criticism of this paper is that functional validation of the proposed mechanism of PMD-dependent immune gene silencing has not been performed. In addition, the authors have not demonstrated that the proposed mechanism of immune evasion (silencing of the HLA locus primarily) is operative in the immunotherapy cohorts they include - are HLA genes silenced in hypomethylated and checkpoint-unresponsive tumours?

It is difficult to perform functional validation because our results suggest that methylation loss and PMD CGI hypermethylation occur globally as a result of cell proliferation. This is a probabilistic, but not mechanistic, event that can be examined better on the correlation basis by large data mining than by validation experiments for individual genes. Instead of performing functional validation, we substantially increased the size of the lung cancer cohort from 40 to 141 patients and performed RNA-seq on 70% of the original cohort samples (n=27/40). We also increased the melanoma cohort size from 15 to 40 by including samples treated with other types of immunotherapy (e.g., interferon). On the other hand, we classified PMDs into three subclasses and found enrichment of immune genes in the class of weaker PMDs (Revised Figure 4). The weak PMDs were accompanied with CGI hypermethylation and gene repression in the pan-cancer data (Revised Figure 4). Eight HLA genes, including *HLA-DQA1*, *HLA-DRA*, and *HLA-DRB1*, were located within the weak PMDs. In response to the reviewer's suggestion, we examined whether the same pattern can be observed in the cohort data. First, we found that CGI promoters in the weak PMDs were actually hypermethylated in the lung cancer and

melanoma cohort samples (Revised Figure 5D and Figure S7A). Moreover, we examined the transcriptome data of our cohort and melanoma cohort. As expected, the genes that were underexpressed in globally demethylated samples were enriched for immune-response pathways including 'MAJOR HISTOCOMPATIBILITY COMPLEX' and 'CYTOKINE-CYTOKINE RECEPTOR INTERACTION' in both cohorts. Moreover, we confirmed that leading edge genes (i.e., core genes that account for enrichment) included HLA genes in the weak PMDs in both cohorts. Although we cannot directly check methylation levels at the CGI promoters of HLA genes due to the absence of methylation probes in these regions, our results indicate the silencing of HLA genes according to global hypomethylation in cohort samples.

We add these results to our revised manuscript as follows, including Supplementary Fig. 6 for the downregulation of HLA genes.

(Page 7, Line 11) Strikingly, immunomodulatory pathway genes involved in antigen processing and presentation, cytokine-cytokine receptor interaction, and JAK-STAT signaling pathway were concentrated in the weak PMDs (**Fig. 4C-D** and **table S4**). The INF- α family genes were in the weak PMDs (**Fig. 4E**). Also, 8 HLA genes, including HLA-DQA1, HLA-DRA, and HLA-DRB1, were located within the weak PMDs.

(Page 7, Line 32) Transcriptome data of the SMC cohort and TCGA cohort showed that genes involved in the major histocompatibility complex and cytokine-cytokine receptor interaction were significantly enriched for repression in globally demethylated tumours of both cohorts (**fig. S6** and **table S6**).

2. The first two figures report on correlations alone- many of which have been described (connection between SCNA and methylation). Also, the authors assume an average methylation level for the tumour types- ignoring that many tumours show varying degrees of hypomethylation. It would be good to consider variability of methylation levels within tumours. - The analyses presented in figures 1a-1d are interesting. However, there are numerous confounders between different tumour types that would be difficult to control for. Do these correlations hold up within single tumour types? For example, if melanoma samples with low vs high global methylation levels are compared, is there still a correlation with CD8 infiltration, etc.?

We reported correlations across many samples within each tumour type below the scatterplots in Figures 1A-D using partial correlation measures adjusting for tumour purity (highlighted by red boxes in the following figure). There, you can see that the sample-wise correlations hold true consistently across different tumour types.

3. The authors state "to focus on the repression of immune-response genes in tumour cells, we excluded genes that are specifically expressed in the immune system from the following analyses." To argue for a tumour cell intrinsic mechanism from bulk data that connects

hypomethylation with repression, looking at genes specifically expressed in the tumour would be preferable.

We think that this is a great suggestion in principle. However, it is not straightforward to define a set of genes that are specifically expressed in tumour. First, there is a great deal of heterogeneity and variability among different cancer types and among individual samples. It does not seem to be possible to select a set of genes that are expressed in common across all different tumours, except for genes involved in cell cycle and proliferation. Second, we are looking at genes that are involved in the immune pathway, not in cell cycle. There is no clear prior knowledge on what kinds of immune response-related genes are expressed commonly in tumours. Such knowledge cannot be gained from cell lines because of the absence of the tumour microenvironment. This is why it is reasonable to exclude genes known to be specifically expressed in the immune cells from gene expression data for tumour tissues.

4. Figure 3 looks at early and late replicating regions based on cell line data and then explores expression of genes in these regions, extrapolating that since PMDs/tumour hypomethylation are reported to be correlated, expression of genes in these late replicating regions would reflect hypomethylation. A better strategy would be to look at expression levels in regions that are hypomethylated, rather than use a different measure (replication timing) in a different system (cell lines) to evaluate expression trends due to hypomethylation.

We thank the reviewer for this valuable comment. In response to the reviewer's suggestion, we explored the expression and promoter methylation of genes residing in hypomethylation domains. Specifically, we identified three distinct subclasses of PMDs based on methylation variability and replication timing. We used methylation variability because PMDs could be characterized and defined by the high variability of methylation levels across samples (Zhou et al. PMID:29610480). Among the three subclasses, immune genes were significantly enriched in the class of weak PMDs that were characterized by relatively lower variability, earlier replication timing, and smaller domain size than the other two subclasses.

From there, we directly show that the PMD genes have CGI promoters that are hypermethylated and thus are repressed in globally demethylated tumours in our revised Fig. 4F and Fig. 4G. In the weak PMDs (Fig. 4F), the number of hypermethylated CpG Island promoters were significantly higher in tumours with low global methylation levels than in tumours with high global methylation levels. The weak PMD genes were indeed underexpressed in globally demethylated tumours (Fig. 4G).

<Hypermethylation of CGI promoters in the weak PMDs in demethylated tumours>

<Repression of genes in the weak PMDs in demethylated tumours>

5. The authors argue that their finding of repression in late replicating regions of demethylated tumours is in contrast to studies showing de-methylating agents upregulate IFNalpha/beta response genes but fail to consider the effect these agents have on island hypermethylation (which is a putative mechanism for upregulation of genes in this experimental setting).

In fact, we did consider the effect of de-methylating agents on CGI hypermethylation in the Discussion as follows.

(Page 14, Line 9) Therefore, different action mechanisms of epigenetic therapy are required when targeting these tumours. Specifically, it needs to be tested whether methylation inhibitors or other epigenetic modulators are capable of restoring the IFN- α/β response and other immunomodulatory pathways by diminishing CGI methylation or loosening heterochromatin structure in these intrinsically demethylated tumours. Our study sheds new light on the combination of epigenetic modulation and checkpoint blockade as a potential precision immunotherapy regimen.

6. Figure 4 looks at PMD methylation as a proxy of overall hypomethylation- this is consistent with previous data that have shown PMDs undergo hypomethylation in tumours; the novelty of this point is weak.

We agree with the reviewer's opinion. In our revision, we do not use PMD methylation levels but only global methylation levels measured from LINE-1 probes.

7. An intriguing finding in this paper is that the survival upon immune checkpoint blockade differs between more and less hypomethylated tumours. The authors should expand their analysis more to evaluate this in other tumours given the relatively small sample size in their cohort. It would also be interesting to evaluate this in other tumour types. Finally, they should

control for stage, age, genotype, etc. Do the tumours with longer survival downregulate HLA genes?

We made substantial efforts to increase sample size. By following the workflow shown below, we were able to collect 60 high-quality DNA samples that passed quality control for both methylome and exome analyses from non-small cell lung carcinoma (NSCLC) patients who were treated with anti-PD-1/PD-L1. We named our cohort after Samsung Medical Center (SMC).

We attempted to expand our analysis further by including an additional cohort. Our thorough literature search led us to an anti-PD-1 NSCLC cohort composed of 81 methylomes and 22 mutanomes (Duruiseaux et al. PMID:30100403) from IDIBELL (Bellvitge Biomedical Research Institute). For another tumour type, we analyzed a TCGA melanoma cohort consisting of 40 samples treated with immunotherapies. The following table is the summary of the resulting cohorts.

Cancer type	Cohort	Number of samples		
		Methylation level	Mutation burden	Aneuploidy level
Lung cancer	SMC	60	60	60
	IDIBELL	81	22	NA
	Combined	141	82	60
Melanoma	TCGA	40	40	40

The following are analysis results for the SMC cohort:

The following are analysis results for the IDIBELL cohort:

The following are analysis results for the TCGA melanoma cohort:

It is notable that the three independent cohort data agree on the importance of global methylation, but not mutation burden.

The following are analysis results for the combined NSCLC cohort:

As suggested by the reviewer, we controlled for tumour state and age (genotype information was not available) in our multivariate analysis of the combined cohort (n = 82) for methylation and mutation burden. As shown below, only global methylation, but not mutation burden, was a significant contributor to patient survival (left) or clinical benefit (right) independently of tumour stage and age.

It should be also emphasized that the results of molecular-level analyses shown in Figure 2 were derived from the following multivariate regression across thousands of pan-cancer samples.

mRNA expression of gene Y ~ β_1 * global methylation level + β_2 * mutation burden + β_3 * aneuploidy level + β_4 * tumour purity + β_5 * age + β_6 * tumour stage

In this manner, we were able to determine that immune infiltrates are associated with global methylation levels independently of mutation burden and aneuploidy when purity, age, and tumour stage are adjusted (Figure 2A). Significant correlations with genomic demethylation were observed also for immunomodulatory pathways that should include genes expressed in tumour cells, independently of mutation burden and aneuploidy (Figure 2B). These include antigen processing and presentation, MHC, cytokine-cytokine receptor interaction, interferon or other cytokine signaling, and complement and coagulation.

8. Figure 7 looks at correlations between SCNA and hypomethylation- which has previously been described. They argue their results show that immune avoidance signatures of highly aneuploid tumours is explained by demethylation- but the language needs to be precise here. It is associated with demethylation but on their analysis alone, it is not possible to argue that it is explained by this (ie. Causation is not proven).

We thank the reviewer for this constructive comment and have toned down our language where necessary as follows:

(Abstract) We also found that genomic hypomethylation ~~explains~~ correlates with the immune escape signatures of aneuploid tumours.

(Page 12, Line 10) These results indicate that the immune avoidance signatures of highly aneuploid tumours ~~can be explained by~~ are associated with genomic demethylation. Indeed, a recent molecular mechanism study⁴¹ contradicted the previous report¹⁰ by suggesting that aneuploid cells generate pro-inflammatory signals for their own elimination by the immune system as a means for cancer cell immunosurveillance.

Reviewer #3 (Remarks to the Author):

In this manuscript, Jung et al., investigate the relationship between global DNA methylation levels and immunological parameters as well as response to immunotherapy using TCGA data and a small lung cancer data-set (n=40). They conclude that global DNA methylation levels is a predictive marker in immunotherapy and imply that epigenetic therapy or cell cycle inhibition could be used in combination with immunotherapy.

I have several major concerns regarding this manuscript.

1. It needs some revision to increase clarity. The use of a professional English editor could help improve clarity and readability.

Our original manuscript has undergone editing by a native speaker. According to the reviewer's suggestion, we will send our manuscript for professional English editing once our revision has gone through the second round of review and reached a decision for minor revision.

2. The global methylation score was based on the array probes that map to LINE1 elements. They validated this score using WGBS for 18 samples that have WGBS and array data available. However, for the WGBS, it was used only CpGs that map to LINE1 elements. Therefore, it was not validated the global methylation levels, only validated the global LINE1 methylation level. It would make more sense to compare the methylation score against all the CpGs in the WGBS data. Alternatively, the authors should not call it 'global methylation level' but rather 'global LINE1 methylation levels'.

Following the reviewer's suggestion, we've changed the term for our measurement to 'global L1 methylation level' throughout the manuscript.

3. Little data is provided on how all the methylation data was normalized from TCGA. Each TCGA cancer type was profiled separately and batch correction is necessary for PAN cancer analysis.

We obtained the batch-corrected and normalized methylation matrix from PanCanAtlas data portal (<https://gdc.cancer.gov/about-data/publications/pancanatlas>). Because we performed all the analyses within each tumour type, we did not adjust for batch effects across different tumour types. We added more description to the Methods section as follows.

(Page 24, Line 4) ~~The processed DNA methylation (based on Infinium Methylation 450k technology),~~ The batch-corrected and normalized DNA methylation data based on Infinium

Methylation 450k technology), together with mRNA expression and gene mutation data, generated by the PanCancer Atlas consortium were obtained from the publication page (<https://gdc.cancer.gov/about-data/publications/pancanatlas>).

4. The global methylation level will be modified by the amount of tumour purity and amount of different immune cells per tumours. Did the authors correct for purity and for the different immune infiltrates? I suggest to use ABSOLUTE to calculate purity and MethyCYBERSORT to calculate the immune-infiltrate from the methylation data. Then correct for this when do the global analysis.

All of our statistical analyses were controlled for tumour purity estimated by ABOLUSTE. For example, we measured partial correlations between global L1 methylation levels and various types of genomic features (mutation burden, aneuploidy level, and the degree of proliferation and immune cell infiltration) while controlling for tumour purity (Figure 1 and Figure S1).

Likewise, the regression models were constructed by adjusting for tumour purity (Figure 2). See the below figure, too, for an additional analysis regarding tumour purity.

Because the immune infiltration level is one of the response variables in our global analyses (Figure 2A), our model cannot be corrected for immune infiltration levels. Instead, we performed the same analysis as Taylor et al. (PMID: 29622463) did. They showed that the correlation between SCNA and immune gene expression disappears when adding the leukocyte fraction to their regression model. Likewise, we attempted to measure the correlation between global L1 methylation and immune gene-set expression after adding the leukocyte fraction to the regression model. We first calculated the leukocyte fraction based on gene expression and methylation data using ESTIMATE (Yoshihara et al. PMID:24113773) and MethyCIBERSORT (Chakravarthy, PMID: 30104673), respectively. As seen below, whereas the addition of the leukocyte fraction resulted in the loss of enrichment for immune genes, the correlation between global L1 methylation and proliferation gene expression maintained. These results suggest that the global methylation level itself is not affected by the leukocyte fraction, which correlates only with the expression of genes supposed to be expressed in immune cells as expected.

5. For instance, the correlation between DNA methylation levels and tumour-infiltrating CD8+ T cells (Fig 1D), could suggest that global tumour hypomethylation is correlated with CD8 infiltration (authors conclusion) or, alternatively, that CD8 T cells have global hypomethylated DNA, and the more CD8 T cells present in the biopsied sample, more demethylation will be observed in the bulk data. This needs to be addressed.

As we responded to the previous comment, our global L1 methylation measure itself is not affected by the leukocyte infiltration, particularly given the remaining association between low global methylation and high proliferation gene expression when leukocyte fraction was controlled for. Biologically speaking, global hypomethylation only occurs in tumours.

6. For the lung cohort, how the 'normal' and 'aberrant' methylation samples were selected? What probes were used? How normal and aberrant was defined? How each sample was assigned to each group?

We apologize for this lack of clarity. In our original submission, we performed hierarchical clustering based on the most variable CpG sites across the cohort samples. This resulted in two visible clusters with distinct patterns on the CGI/shore and open sea. We arbitrarily defined the normal and aberrant methylome clusters by examining this pattern. For clarity, in our revised manuscript, the samples were divided into two groups according to the median of the global L1

methylation levels. We annotated the two groups as 'low' (< median) and 'high' (\geq median) instead of referring them to 'normal' and 'aberrant'. We added the description in the figure legends and methods of our revised manuscript. We attach the resulting heatmap below.

(Page 7, Line 27) ~~The samples from the combined lung cancer cohort were divided into global low versus high methylation groups according to the L1 methylation levels.~~

7. I disagree with the authors conclusions about cell cycle inhibitors in the discussion. These patients already have the DNA methylation change before treatment. Cell cycle inhibitors will not prevent or revert these changes, even if the demethylation was initially caused by cell divisions.

We agree with the reviewer's opinion so that we have removed the related statements in the abstract and discussion as follows while leaving the statements regarding the interpretation of the reported effects of cell cycle inhibition.

Discussion:

(Page 13, Line 19) There are multiple studies that reported antitumour immunity augmented by CDK4/6 inhibition and synergistic effects of the cell cycle inhibitors and checkpoint blockade⁴¹⁻⁴³. Our results suggest that cell cycle inhibition may bring about opposing effects by suppressing genetic alterations that facilitate neoantigen formation and at the same time, preventing immune evasion promoted by epigenetic alterations that repress immunomodulatory pathway genes. Hence, the reported effects of cell cycle inhibition suggest that the benefits achieved by epigenetic influences may be greater than the adverse effects caused by suppressing neoantigen formation. ~~From the perspective of precision immunotherapy, tumours with active cell proliferation and global methylation loss may particularly benefit from the augmented treatments based on cell cycle inhibition.~~

Abstract:

Hence, DNA methylation alternations implicate epigenetic modulation ~~and cell cycle inhibition~~ as a combination regimens for precision immunotherapy.

8. Reference #28 and #44 are the same.

Corrected.

Reviewers' Comments:

Reviewer #1:

Remarks to the Author:

My biggest concern (shared with other reviewers) had to do with the sample sizes used to compare associations between methylation, TMB, and scna. The authors significantly increased their sample sizes by incorporating data from additional studies, and I am satisfied that the original results have been sufficiently validated. The authors also adequately addressed my other concerns, and thus this article is nearly suitable for publication in my opinion. However, in the process of this revision, they made substantial change to how they have defined the "PMD" regions in several analyses, and this raised some additional concerns on my part. I believe they are addressable, but may require further refinements to the analysis. I would appreciate responses to the questions below and any necessary revisions before I would want to give my full endorsement of this manuscript.

The "weak PMDs" you describe would be a new concept in the field, so it deserves some scrutiny in order to avoid proliferation of new terms for existing concepts. I am struck by the fact that a large number of weak PMDs fall in single 100kb bins (your Figure 4b). It appears that the 100kb bins you are using have arbitrary tiled boundaries (starting at 0kb, 100kb, 200kb, etc. along the genome). So I'm wondering whether the low variability and earlier replication timing you observe could be caused by many/most of these weak PMD bins overlapping an HMD/PMD boundary. If this were the case, only a portion of the 100kb bin would fall within the PMD side, and thus your variability and replication timing estimates would be at intermediate levels between the HMD and PMD sides. This could be easily tested by checking the rate at which weak PMDs are adjacent to HMDs, compared moderate and strong PMD bins. If many more weak PMDs were adjacent to HMDs, it would suggest a boundary effect. This would actually be quite interesting and quite consistent with Berman et al. 2012. PMID: 22120008, which showed that hypermethylated CpG islands were concentrated at internal boundaries of PMDs within about 300kb from the HMD/PMD boundary (Figure 5C from Berman, 2012). These internal boundary regions are thus expected to contain more silenced genes than regions farther away from the boundary, which might explain the overabundance of silenced immune genes within these regions. If this turns out to be the case, I think it would be more straightforward to come up with a name like "PMD internal boundaries" , and cite previous studies, rather than coming up with a new concept of "weak PMDs". It would also be important to confirm that the enriched classes of immune genes actually fall within the PMD side (I expect this to be the case since hypermethylated CpG islands almost always fall on the PMD side in Figure 5C from Berman, 2012).

The IFN chr9p22 complex shown in Figure 4e is a somewhat conflicting example of this. On the one hand, it does fall at an internal PMD mostly within 300kb of the HMD/PMD boundary so it would fit that definition. However, in the prior version of this figure, most of the cluster was categorized as "common PMD", which has the highest degree of variability, so it is maybe not typical of most "Weak PMDs" which have lower variability levels.

Reviewer #2:

Remarks to the Author:

The authors have responded and largely addressed prior criticisms.

Reviewer #3:

Remarks to the Author:

The authors have addressed all my concerns.

Reviewers' comments:

Reviewer #1 (Remarks to the Author):

My biggest concern (shared with other reviewers) had to do with the sample sizes used to compare associations between methylation, TMB, and scna. The authors significantly increased their sample sizes by incorporating data from additional studies, and I am satisfied that the original results have been sufficiently validated. The authors also adequately addressed my other concerns, and thus this article is nearly suitable for publication in my opinion. However, in the process of this revision, they made substantial change to how they have defined the "PMD" regions in several analyses, and this raised some additional concerns on my part. I believe they are addressable, but may require further refinements to the analysis. I would appreciate responses to the questions below and any necessary revisions before I would want to give my full endorsement of this manuscript.

The "weak PMDs" you describe would be a new concept in the field, so it deserves some scrutiny in order to avoid proliferation of new terms for existing concepts. I am struck by the fact that a large number of weak PMDs fall in single 100kb bins (your Figure 4b). It appears that the 100kb bins you are using have arbitrary tiled boundaries (starting at 0kb, 100kb, 200kb, etc. along the genome). So I'm wondering whether the low variability and earlier replication timing you observe could be caused by many/most of these weak PMD bins overlapping an HMD/PMD boundary. If this were the case, only a portion of the 100kb bin would fall within the PMD side, and thus your variability and replication timing estimates would be at intermediate levels between the HMD and PMD sides. This could be easily tested by checking the rate at which weak PMDs are adjacent to HMDs, compared moderate and strong PMD bins. If many more weak PMDs were adjacent to HMDs, it would suggest a boundary effect. This would actually be quite interesting and quite consistent with Berman et al. 2012. PMID: 22120008, which showed that hypermethylated CpG islands were concentrated at internal boundaries of PMDs within about 300kb from the HMD/PMD boundary (Figure 5C from Berman, 2012). These internal boundary regions are thus expected to contain more silenced genes than regions farther away from the boundary, which might explain the overabundance of silenced immune genes within these regions. If this turns out to be the case, I think it would be more straightforward to come up with a name like "PMD internal boundaries", and cite previous studies, rather than coming up with a new concept of "weak PMDs". It would also be important to confirm that the enriched classes of immune genes actually fall within the PMD side (I expect this to be the case since hypermethylated CpG islands almost always fall on the PMD side in Figure 5C from Berman, 2012). The IFN chr9p22 complex shown in Figure 4e is a somewhat conflicting example of this. On the one hand, it does fall at an internal PMD mostly within 300kb of the HMD/PMD boundary so it would fit that definition. However, in the prior version of this figure, most of the cluster was categorized as "common PMD", which has the highest degree of variability, so it is maybe not typical of most "Weak PMDs" which have lower variability levels.

We thank the reviewer for these valuable comments. As pointed out by the reviewer, the 100 kb bins have arbitrary tiled boundaries, and each bin has methylation variability value defined by the previous study (Zhou et al. PMID:29610480). According to this value, each bin was classified into a PMD (>0.125) or HMD (< 0.125 ; the cutoff value was defined by Zhou et al). In order to identify PMDs reliably, we merged consecutive PMDs and retained those > 300 kb in length and assigned the average methylation variability value into each merged PMD for clustering analysis (Figure 4a-b; see methods). Therefore, the length of

PMDs correlates with their distance to HMDs because the merged PMDs are surrounded by HMDs.

We need to point out that the concept of weak PMDs has already been introduced (Salhab et al, PMID:30266094). In this work, short PMDs were characterized by high gene density and early replication timing (i.e, early/mid S phase). Therefore, instead of introducing a new concept in the field, we can borrow the concept of short PMDs by citing this paper. We have now revised our manuscript and Figure 4 accordingly.

In response to the reviewer's insightful comments, we sought to test whether the "PMD internal boundary" effect is able to fit our observation for the overabundance of the silenced immune genes. We examined the extent to which the immune genes fall near PMD boundaries. For a total of 77 unique immune genes in the pathways enriched for short PMDs (n=13; Figure 4c), we calculated the average distance to the nearest HMD, which turned out to be 143kb, and then estimated its P value by generating a background distribution. We randomly picked 77 genes in PMDs (i.e., random gene set) and calculated the average distance of them to the nearest HMD. This procedure was repeated 10,000 times. As a result, we found that the observed distance (143kb) is significantly shorter than expected from background ($P = 0$). This result, provided as Figure 4H in our revised manuscript, supports immune gene silencing by the CGI hypermethylation of PMD boundaries. This is consistent with the report that hypermethylated CGIs are concentrated within about 300kb from the boundary (Berman et al. 2012. PMID:22120008).

At this point, we need to make it clear that the boundary effect cannot explain the depletion of immune genes near the side of intermediate and long PMDs. The conclusion that we can draw from all the data and report is that immune genes are overrepresented in short PMDs with early replication timing and low methylation variability (our previous finding) and are concentrated near PMD boundaries (finding thanks to the reviewer's comments), and that silencing of immune genes can be explained by the CGI hypermethylation of PMD boundaries (Berman et al. 2012. PMID:22120008).

As the reviewer pointed out, our previous IFN chr9p22 figure was covered by 300kb PMDs where common PMD and PMD accounted for 200kb and 100kb of them, respectively. According to Zhou's cutoff value, we previously classified bins for which methylation variability value is greater than 0.15 as common PMDs. After merging PMDs as we mentioned above, the IFN chr9p22 region belongs to the merged PMD for which length and average methylation variability value are 700kb and 0.1587, respectively. Our clustering analysis based on replication timing and methylation variability classified this region as

short PMD, meaning that this region has relatively earlier replication timing and lower methylation variability compared to those belonging to intermediate or long PMDs. One of our contributions was to reveal a distinctive class of common PMD (shorter size, earlier replicated, and lower variability), which is different from conventional common PMDs defined by Zhou's arbitrary cutoff (0.15).

We have now revised our manuscript and Figure 4 according to these additional findings (attached below). We thank the reviewer again for these insightful comments and hope that our response will be sufficient for your full endorsement of this manuscript.

PMDs were characterized by and defined based on the high variability of solo-WCGW methylation levels across samples¹⁶. Our inspection of the methylation variability and replication timing of various PMDs led to three distinct subclasses (**Fig. 4A**). **In accordance with a previous report³⁶, the properties of PMDs were associated with their genomic length with shorter PMDs characterized by earlier replication timing (Fig. 4A-B). Strong PMDs were characterized by high variability, later replication timing, and large domain size whereas weak PMDs had low variability in methylation, relatively earlier replication timing, and small domain size (Fig. 4A-B).** Strikingly, immunomodulatory pathway genes involved in antigen processing and presentation, cytokine-cytokine receptor interaction, and JAK-STAT signaling pathway were concentrated in the **short** PMDs (**Fig. 4C-D** and **table S4**). The INF- α family genes were in the **short** PMDs (**Fig. 4E**). Also, 8 HLA genes, including HLA-DQA1, HLA-DRA, and HLA-DRB1, were located within the **short** PMDs. Consistent with the late-replicating regions (**Fig. 3A-B**), the **short** PMDs were accompanied with gene repression (**Fig. 4F**) and CGI hypermethylation (**Fig. 4G**) in demethylated tumours. **Hypermethylated CGIs are most abundant within 150 kb of PMD boundaries³⁷. The enriched immune genes (Fig. 4C-D) were significantly concentrated near PMD boundaries with the average distance of 143 kb (Fig. 4H), suggesting that these genes are particularly prone to promoter methylation.**

Figure 4

Reviewers' Comments:

Reviewer #1:

Remarks to the Author:

The authors have now addressed all my concerns.